# Learning Physics-Grounded 4D Dynamics with Neural Gaussian Force Fields

**Shiqian Li** [1,2,6,7]*  **Ruihong Shen** [3,2,6,7]*  **Junfeng Ni** [4]  **Chang Pan** [1,2,5,6,7]
**Chi Zhang** [1,6] ✉  **Yixin Zhu** [2,1,6,7] ✉

*Equal contribution   ✉ Corresponding author

[1] Institute for AI, Peking University   [2] School of Psychological and Cognitive Sciences, Peking University
[3] School of EECS, Peking University   [4] Department of Automation, Tsinghua University
[5] Yuanpei College, Peking University   [6] State Key Lab of General AI, Peking University
[7] Beijing Key Laboratory of Behavior and Mental Health, Peking University
https://neuralgaussianforcefield.github.io/

## Abstract

Predicting physical dynamics from visual data remains a fundamental challenge in Artificial Intelligence (AI), as it requires both accurate scene understanding and robust physics reasoning. While recent video generation models achieve impressive visual quality, they lack explicit physics modeling and frequently violate fundamental laws like gravity and object permanence. Existing approaches combining 3D Gaussian splatting with traditional physics engines achieve physical consistency but suffer from prohibitive computational costs and struggle with complex real-world multi-object interactions. The key challenge lies in developing a unified framework that learns physics-grounded representations directly from visual observations while maintaining computational efficiency and generalization capability. Here we introduce Neural Gaussian Force Field (NGFF), an end-to-end neural framework that learns explicit force fields from 3D Gaussian representations to generate **interactive, physically realistic 4D videos** from multi-view RGB inputs, achieving two orders of magnitude speedup over prior Gaussian simulators. Through explicit force field modeling, NGFF demonstrates superior spatial, temporal, and compositional generalization compared to state-of-the-art (SOTA) methods, including Veo3 and NVIDIA Cosmos, while enabling robust sim-to-real transfer. Comprehensive evaluation on our GSCollision dataset—640k rendered physical videos (~4TB) spanning diverse materials and complex multi-object interactions—validates NGFF's effectiveness across challenging scenarios. Our results demonstrate that NGFF provides an effective bridge between visual perception and physical understanding, advancing video prediction toward physics-grounded world models with interactive capabilities.

## 1 Introduction

From infancy, humans develop robust intuitive physics that enables rapid inference of object properties and dynamics from visual input (Spelke & Kinzler, 2007; Battaglia et al., 2013; Piloto et al., 2022; Pramod et al., 2025). This remarkable ability allows us to predict how objects will move, collide, and deform in complex 3D environments using our internal "physics engine" (Battaglia et al., 2013; Kubricht et al., 2016; Ullman et al., 2017; Wang et al., 2024; Bi et al., 2025).

Current AI systems fall far short of this capability (Bansal et al., 2024; Bordes et al., 2025). While recent video generation models produce visually impressive results and show promise as "world simulators" (Ho et al., 2022; Yang et al., 2024; Yu et al., 2025; Wiedemer et al., 2025), they fundamentally lack physical understanding. These models frequently violate basic principles like object permanence, solidity, and gravity, even after training on millions of videos (Kang et al., 2025; Duan et al., 2025; Chow et al., 2025; Li et al., 2025a; Motamed et al., 2025). This limitation severely constrains AI agents' ability to interact effectively with real-world physical environments.

Achieving human-level physical reasoning in AI faces two fundamental challenges. **First, learning effective object representations from RGB inputs.** Most physics prediction methods rely on precise object-centric data (Rubanova et al., 2024; Ma et al., 2023; Li et al., 2025b) or implicit volumetric encodings that are difficult to ground in physics (Driess et al., 2023; Xue et al.,

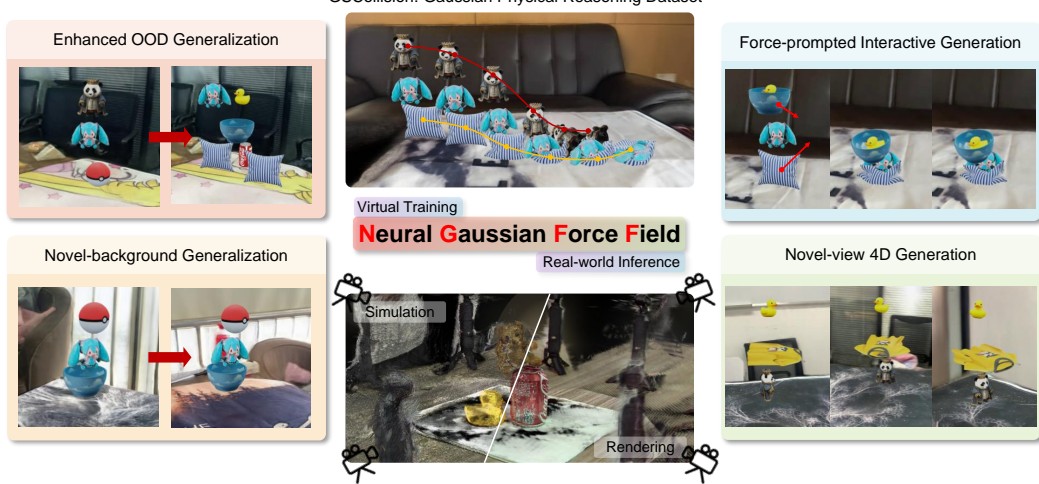

Figure 1: **Capabilities of NGFF.** NGFF is a physics-grounded video prediction framework that unifies perception and dynamics to model complex interactions and synthesize 4D videos. Built on Gaussian representations and force fields, it enables novel-view and novel-background synthesis as well as force-prompted interactive generation (Section 4.3). Moreover, NGFF achieves strong spatial and temporal generalization in dynamic prediction (Section 4.2) and can be effectively adapted to real-world scenarios (Section 4.4).

2023b). Particle-based approaches (Li et al., 2019; Allen et al., 2023b; Whitney et al., 2024) require additional pretrained renderers for multi-view consistency, limiting scalability to complex scenes. **Second, learning generalizable physical dynamics.** Large video models tend to overfit superficial visual features and retrieve training patterns rather than learning robust physical principles (Kang et al., 2025; Li et al., 2022; Shiri et al., 2024). Recent Gaussian splatting methods for physics (Xie et al., 2024; Lin et al., 2025b; Jiang et al., 2025a; Zhang et al., 2025) show promise but struggle with scalability and generalization due to predefined simulators and intractable parameters.

We introduce Neural Gaussian Force Field (NGFF), a physics-grounded neural framework that addresses these challenges through explicit force field modeling. NGFF encodes multi-view RGB images into 3D Gaussian representations with object semantics via a feed-forward geometry transformer (Wang et al., 2025a;b). A neural operator then predicts object-centric force fields, which are integrated through an Ordinary Differential Equation (ODE) solver to simulate realistic dynamics. The framework renders the evolved Gaussians to generate physically consistent multi-view videos.

NGFF demonstrates four key capabilities (Figure 1): **enhanced Out-of-Distribution (OOD) generalization** through explicit force field reasoning that enables robust prediction across complex interactions; **interactive generation** via force-prompted control of learned dynamics; **efficient multi-view synthesis** with background-agnostic video generation from object-aware 3D Gaussians; and **sim-to-real transfer** through neural field representations that generalize to real-world scenarios.

To support training and evaluation, we construct GSCollision, a comprehensive 4D Gaussian dataset featuring diverse rigid and soft body physics across phenomena including falling, collision, rotation, sliding, and containment. The dataset incorporates real-world backgrounds from WildRGBD (Xia et al., 2024) to enhance visual complexity and realism, totaling 640k rendered videos (~4TB).

We evaluate NGFF across dynamic prediction, video generation, and real-world transfer scenarios. Results demonstrate that our method generates high-quality predictive videos while achieving physically plausible simulations in unseen scenarios. NGFF surpasses SOTA particle-based methods like Pointformer (Wu et al., 2024b) and video generation models including Veo3 (DeepMind, 2025), NVIDIA Cosmos (NVIDIA et al., 2025), and PhysGen3D (Chen et al., 2025). Our work bridges the gap in Gaussian-based simulation (Zhang et al., 2025; Jiang et al., 2025a; Zhobro et al., 2025) by simultaneously capturing high visual complexity and complex multi-object physical interactions.

## 2 RELATED WORK

Physical reasoning and visual dynamic prediction are fundamental challenges in developing AI systems with human-like intuitive physics. Evaluation frameworks based on the Violation-of-

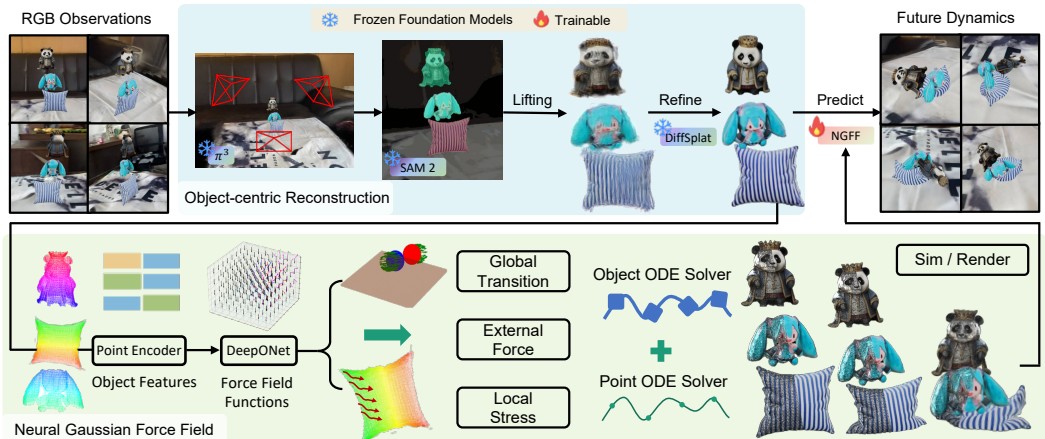

Figure 2: **Overall framework of NGFF.** Given unposed RGB inputs, our approach first reconstructs the scene into object-aware 3D Gaussians through feed-forward prediction, followed by segmentation and refinement to handle occlusions and noise. The refined Gaussians are encoded into high-dimensional features and processed by a DeepONet-based neural operator to predict object-centric force fields. These force fields are integrated through ODE solvers to simulate realistic dynamics, enabling iterative prediction and rendering of future scene states with maintained physical consistency.

Expectation (VoE) paradigm (Piloto et al., 2022; Dai et al., 2023; Bakhtin et al., 2019; Allen et al., 2020; Bear et al., 2021; Li et al., 2024) have driven progress in neural dynamics prediction, where Graph Neural Networks (GNN)-based approaches typically employ mesh, SDF, spring-mass, or particle-grid representations (Sanchez-Gonzalez et al., 2020; Bear et al., 2021; Allen et al., 2023a; Rubanova et al., 2024; Jiang et al., 2025a; Zhang et al., 2025). While these methods succeed in simulating diverse materials, they struggle with generalization to complex, out-of-distribution interactions due to reliance on predefined physical models and structured inputs. Parallel efforts in differentiable physics simulators, particularly MPM-based Gaussian formulations (Xie et al., 2024; Lin et al., 2025b; Chen et al., 2025), achieve high physical fidelity but suffer from prohibitive computational costs that limit real-world scalability. Scene representations have evolved from point clouds and NeRFs (Shi et al., 2024; Whitney et al., 2024; Driess et al., 2023; Xue et al., 2023a; Li et al., 2023) to 3D Gaussian Splatting (Kerbl et al., 2023), with recent advances enabling rapid reconstruction through feed-forward prediction (Wang et al., 2025a; Jiang et al., 2025b; Wang et al., 2025b; Zhobro et al., 2025). Our work unifies feed-forward Gaussian-based scene representations with neural dynamics modeling to enable generalizable physical reasoning across multi-object interactions, learning physics directly from visual observations while maintaining computational efficiency and generalization capability. Detailed related work is provided in Section D.

## 3 METHOD

We formulate 4D video prediction as learning Neural Force Fields (NFFs) that govern the temporal evolution of 3D Gaussian scene representations. Our approach consists of two complementary components: feed-forward reconstruction that converts multi-view RGB observations into object-aware 3D Gaussians, and neural dynamics prediction that simulates realistic physics through learned force fields integrated via ODE solvers (Figure 2). This unified framework enables both accurate scene reconstruction and physically grounded temporal prediction while maintaining computational efficiency for real-world applications.

### 3.1 PROBLEM FORMULATION

Consider a dynamic scene observed through $N$ unposed RGB images $\mathcal{I}_0 = \{I_0(p_k) \in \mathbb{R}^{H \times W \times 3} \mid k = 1, \ldots, N\}$ captured from different viewpoints $p_k \in \mathbb{R}^6$ at the initial time step. Our objective is to predict the scene's temporal evolution, generating future observations $I_t(p)$ at arbitrary time steps $t \in T$ and camera poses $p \in \mathbb{R}^6$ that respect both visual consistency and physical constraints.

We represent scenes using $M$ 3D Gaussians $\mathcal{G}_0 = \{g_{0,i}\}_{i=1}^M$ that encode both geometric and semantic properties extracted from initial observations $\mathcal{I}_0$. The core challenge lies in learning a neural

dynamics model $f_\theta$ that predicts physically plausible state transitions $\mathcal{G}_{t+1} = f_\theta(\mathcal{G}_t)$ while enabling efficient rendering through Gaussian splatting $I_{t,k} = \text{Render}(\mathcal{G}_t, p_k)$.

Our learning framework optimizes dynamics prediction through the objective $\min_\theta \mathcal{L}(\hat{\mathcal{G}}_t, \mathcal{G}_t)$, where $\hat{\mathcal{G}}_t$ is the predicted Gaussian state at time $t$ and $\mathcal{L}$ enforces physical consistency in the predicted Gaussian trajectories.

### 3.2 THE NEURAL GAUSSIAN FORCE FIELD (NGFF) FRAMEWORK

#### 3.2.1 FEED-FORWARD 3D RECONSTRUCTION

**Geometric and appearance reconstruction**  We build upon proven transformer-based architectures (Wang et al., 2025a;b) for robust scene reconstruction from unposed multi-view images. Input images are first tokenized using DINOv2 (Oquab et al., 2024), then processed through an $L$-layer Alternating-Attention Transformer that captures global geometric relationships across all $N$ viewpoints. Three specialized decoder heads predict: (i) camera poses $p$ and (ii) Gaussian centers $\mu$ via pixel-shuffling decoders (Shi et al., 2016), and (iii) Gaussian attributes ($\alpha, \mathbf{r}, \mathbf{s}, \mathbf{c}$) through convolutional upsampling with RGB shortcuts (Ye et al., 2025) to preserve fine-grained appearance details.

**Object-centric reconstruction**  Physics simulation requires decomposing scenes into distinct interacting objects. We leverage SAM2 (Ravi et al., 2025) to generate pixel-wise instance masks, which are back-projected onto reconstructed Gaussians through majority voting to partition $\mathcal{G}$ into $K$ object groups $\mathcal{G}_k = \{g \in \mathcal{G} \mid \text{label}(g) = k\}$. To address occlusions and invisible parts from inputs, we refine object representations using DiffSplat (Lin et al., 2025a) with $\text{Sim}(3)$ pose estimation to enhance topological completeness (details in Section B.2). In refinement, we choose single-view instead of multi-view images as input since it works better for final 3D generation.

#### 3.2.2 ODE-BASED NEURAL DYNAMICS SIMULATOR

Our dynamics model centers on learning **force fields**—vector functions $\mathbf{F}(\cdot)$ that predict forces acting on objects based on their current states. This physics-grounded approach enables unified modeling of rigid and soft body interactions while achieving robust generalization to unseen scenarios through explicit force field modeling.

**Force field prediction**  The core of our framework leverages the physical principle of force fields—vector fields $\mathbf{F}(\cdot)$ that determine forces $\mathbf{F}(\mathbf{z}^q(t))$ acting on query objects $q$ based on their states $\mathbf{z}^q(t)$ at time $t$. We represent each object's state as $\mathbf{z}^q(t) = \{\mathbf{h}^q, \mathbf{s}^q(t), \dot{\mathbf{s}}^q(t)\}$, encoding: (1) **semantic features** $\mathbf{h}^q$: object-level features extracted from Gaussian centers via PointNet (Qi et al., 2017), (2) **zeroth-order states** $\mathbf{s}^q(t)$: local point cloud $\mathbf{x}(t) \in \mathbb{R}^{M \times 3}$, center of mass $\mathbf{c}(t) \in \mathbb{R}^3$, and orientation (Euler angles) $\boldsymbol{\theta}(t) \in \mathbb{R}^3$, and (3) **first-order states** $\dot{\mathbf{s}}^q(t)$: local point cloud velocity $\dot{\mathbf{x}}(t) \in \mathbb{R}^{M \times 3}$, velocity of center of mass $\dot{\mathbf{c}}(t) \in \mathbb{R}^3$, and angular velocity $\dot{\boldsymbol{\theta}}(t) \in \mathbb{R}^3$.

For scenes with $K$ interacting objects, we model the **global transformation force field** $\mathbf{F}^{\text{global}}(\cdot) \in \mathbb{R}^6$—encompassing both translational and rotational components—using a neural operator over a relational graph $\mathcal{N} = (V, E)$, where $V = \{\mathbf{z}^0(t), \ldots, \mathbf{z}^{K-1}(t)\}$ denotes object nodes and $E$ encodes physical contacts. Inspired by neural operator learning (Lu et al., 2021) and relational inductive biases (Battaglia et al., 2018), we define:

$$\mathbf{F}^{\text{global}}(\mathbf{z}^q(t)), \mathbf{F}^{\text{latent}}(\mathbf{z}^q(t)) = \sum_{i \in \mathcal{N}(q)} \mathbf{W} \left( f_\eta(\mathbf{z}^i(t)) \odot f_\phi(\mathbf{z}^q(t)) \right) + \mathbf{b}, \qquad (1)$$

where $\mathcal{N}(q)$ denotes neighboring objects in contact with $q$, $f_\eta$ and $f_\phi$ are neural encoders with learnable parameters, $\odot$ represents element-wise product, and projection matrix $\mathbf{W} \in \mathbb{R}^{d_{\text{hidden}} \times d_{\text{force}}}$ with bias $\mathbf{b} \in \mathbb{R}^{d_{\text{force}}}$ map hidden features to force vectors. This formulation captures diverse interactions, including contact, sliding, and gravity.

To model local deformations in **soft bodies**, we introduce neural network $\Phi$ that predicts point-wise **local stress fields** $\mathbf{F}^{\text{local}}(\cdot) \in \mathbb{R}^{M \times 3}$ based on a **Contact Area Mask** (CAM) highlighting contact regions and $\mathbf{F}^{\text{latent}}$ describing the force distribution on an object:

$$\mathbf{F}^{\text{local}}(\mathbf{z}^q(t)) = \Phi \left( \mathbf{F}^{\text{latent}}(\mathbf{z}^q(t)), \text{CAM}, \mathbf{x}^q(t), \dot{\mathbf{x}}^q(t) \right). \qquad (2)$$

The unified force field combines both components: $\mathbf{F}(\mathbf{z}^q(t)) = \left(\mathbf{F}^{\text{local}}, \mathbf{F}^{\text{global}}\right)$.

$$\mathbf{F}(\mathbf{z}^q(t)) = \left(\mathbf{F}^{\text{local}}, \mathbf{F}^{\text{global}}\right). \tag{3}$$

**Trajectory decoding via ODE integration**  To recover continuous and physically plausible trajectories, we integrate learned force fields using second-order ODE solvers (Chen et al., 2018). Object trajectories are computed as:

$$\mathbf{z}^q(t) = \text{ODESolve}\left(\mathbf{z}^q(0), \mathbf{F}, 0, t\right), \tag{4}$$

$$\mathbf{s}(t) = \mathbf{s}(0) + \int_0^t \dot{\mathbf{s}}(t)\, dt, \quad \dot{\mathbf{s}}(t) = \dot{\mathbf{s}}(0) + \int_0^t \mathbf{F}(\mathbf{z}^q(t))\, dt. \tag{5}$$

This formulation provides a fully differentiable bridge between NFF predictions and physical dynamics simulation.

### 3.2.3  TRAINING STRATEGY

Our training employs a two-stage approach that leverages both real-world visual data and physics simulation ground truth. The feed-forward reconstruction module initializes with pretrained $\pi^3$ parameters, where the feature encoder, point head, and camera head remain frozen while the splatter head is fine-tuned on WildRGBD using combined RGB and geometric consistency losses to align predicted and rendered depth maps. The neural dynamics simulator trains separately on synthetic Material Point Method (MPM) simulation data, optimizing Mean Squared Error (MSE) loss between predicted Gaussian configurations and motion trajectories against ground-truth simulations. This decoupled strategy enables effective utilization of both domains while maintaining training stability.

### 3.3  CAPABILITIES OF THE NEURAL GAUSSIAN FORCE FIELD (NGFF)

**Dynamic prediction as operator learning**  NGFF formulates dynamic prediction as neural operator learning over explicit force fields, providing a unified framework for modeling both rigid and deformable objects within a shared representational space. By employing neural operators on relational graphs, our approach naturally captures complex physical phenomena including contact interactions, collision dynamics, and material deformation. This operator-based formulation enables robust scalability to multi-body systems while achieving strong generalization across spatial configurations, temporal horizons, and compositional variations in object types and arrangements.

**Video generation as efficient rendering of physical trajectories**  Our framework bridges perception and simulation by combining feed-forward 3D Gaussian reconstruction with learned force field dynamics. The resulting system generates videos through differentiable Gaussian splatting that renders predicted trajectories with reasonable photorealistic quality and good physical consistency. This unified approach naturally supports flexible viewpoint synthesis, contextual scene variations, and interactive interventions, enabling applications ranging from novel view synthesis to what-if scenario exploration while maintaining computational efficiency compared to traditional physics simulators.

**Real-world transfer**  The modular design of NGFF facilitates effective sim-to-real transfer through two key mechanisms. First, 3D Gaussians provide a disentangled interface that abstracts noisy visual inputs into clean geometric representations suitable for physics modeling. Second, NFF operators learn robust physical relationships that generalize beyond synthetic training data. This combination enables the framework to adapt learned dynamics to real-world RGB observations while preserving physical consistency, bridging the gap between controlled simulation environments and complex real-world scenarios.

## 4  EXPERIMENTS

### 4.1  DATASET

We introduce GSCollision, a comprehensive 3D Gaussian-splats physical reasoning dataset that advances beyond existing benchmarks (Bakhtin et al., 2019; Bear et al., 2021; Greff et al., 2022; Li

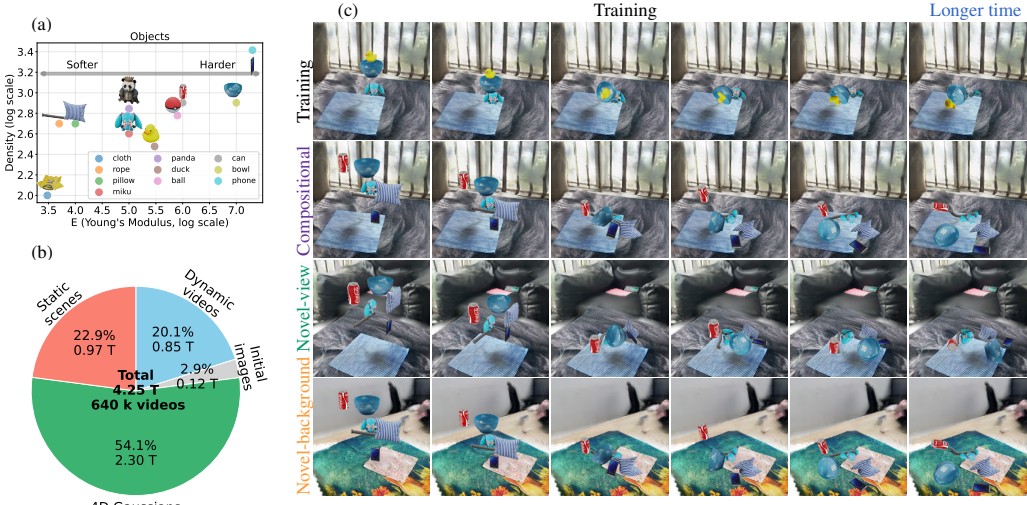

Figure 3: **GSCollision dataset.** (a) Distribution of 10 representative objects characterized by density and material hardness (Young's modulus, log scale). The parameter space spans from soft, lightweight materials (*e.g.*, cloth, rope, pillow) in the lower-left region to rigid, dense objects (*e.g.*, bowl, phone) in the upper-right, providing comprehensive coverage of everyday material properties. (b) Dataset composition totaling 4.25 TB across 3,200 scenes and 640k videos. The pie chart shows storage distribution among training and test splits, multi-view initial scene captures, and auxiliary data files. (c) Representative frame gallery across evaluation scenarios: training sequences, longer temporal rollouts, compositional generalization, novel viewpoints, and novel backgrounds, demonstrating the diversity of physical interactions and visual contexts in our benchmark.

et al., 2024) by providing physics-grounded 3D representations suitable for both perception and dynamics modeling. Constructed using MPM simulators (Xie et al., 2024), our dataset captures realistic behaviors of both rigid and deformable bodies while maintaining the computational efficiency of Gaussian-based representations for fast, differentiable rendering.

The dataset features 10 everyday objects with diverse material properties—ranging from soft items like pillows and ropes to rigid objects such as balls and phones—that exhibit distinct physical behaviors under various interaction scenarios. Through systematic sampling of object compositions and spatial configurations within a 3D environment, we generate 3,200 physically realistic scenarios encompassing object–object and object–ground interactions across diverse dynamics including falling, collisions, rotation, sliding, and containment.

Our evaluation protocol introduces systematic distributional shifts to assess generalization capabilities. The training set contains 2,700 three-object scenes, while the test set comprises 500 scenarios with deliberate complexity variations: 300 unseen three-object configurations for spatial generalization, 100 four-object scenes for compositional scaling, and 100 six-object scenes for complex multi-body reasoning. Each sequence spans 100 simulation steps (approximately two seconds), capturing rich temporal dynamics across scenarios such as stacked towers, container-based interactions, and collision-driven behaviors. Dataset statistics and analysis are provided in Figure 3.

## 4.2 DYNAMIC PREDICTION

We evaluate NGFF's physics modeling capabilities across four critical generalization dimensions that test fundamental aspects of learned dynamics. **Spatial generalization** assesses force field predictions at unseen object positions, requiring accurate spatial extrapolation beyond training configurations. **Temporal generalization** evaluates long-term stability and accuracy over extended rollouts that exceed training sequence lengths. **Compositional generalization** (Lake & Baroni, 2023) probes reasoning about novel object combinations and scaling to larger multi-body systems (4–6 objects) not encountered during training. **External force generalization** tests adaptability under previously unseen perturbations and intervention scenarios.

We benchmark against established SOTA approaches, including particle-based methods (GCN (Kipf & Welling, 2017), Pointformer (Wu et al., 2024b)) and physics-engine baselines using MPM simulators with Vision Language Models (VLM)-estimated parameters (Chen et al., 2025). Complete

Table 1: **Dynamic prediction performance across generalization scenarios.** We evaluate models using four key metrics: Root Mean Squared Error (RMSE) between predicted and ground-truth trajectories for positional accuracy, Final Position Error (FPE) for long-term stability, correlation coefficient R for temporal consistency, and average inference time over 100 simulation steps for computational efficiency. Arrows indicate whether higher (↑) or lower (↓) values represent better performance. The ablation study NGFF w/o deform. demonstrates our framework's performance when soft body deformation modeling is disabled, isolating the contribution of local stress field prediction.

| Model | Spatial | | | Temporal | | | Compositional | | | Time |
|---|---|---|---|---|---|---|---|---|---|---|
| | RMSE ↓ | FPE ↓ | R ↑ | RMSE ↓ | FPE ↓ | R ↑ | RMSE ↓ | FPE ↓ | R ↑ | (s) ↓ |
| VLM-MPM | 0.306 | 0.774 | 0.299 | 0.328 | 0.901 | 0.300 | 0.358 | 0.904 | 0.305 | 39.29 |
| GCN | 0.134 | 0.479 | 0.406 | 0.174 | 0.590 | 0.400 | 0.145 | 0.509 | 0.347 | 0.346 |
| Pointformer | 0.096 | 0.394 | 0.623 | 0.129 | 0.537 | 0.604 | 0.162 | 0.594 | 0.434 | **0.183** |
| NGFF w/o deform. | 0.110 | 0.459 | 0.595 | 0.144 | 0.600 | 0.578 | 0.131 | 0.546 | 0.515 | 0.303 |
| **NGFF** | **0.082** | **0.326** | **0.661** | **0.107** | **0.419** | **0.652** | **0.104** | **0.409** | **0.571** | 0.363 |

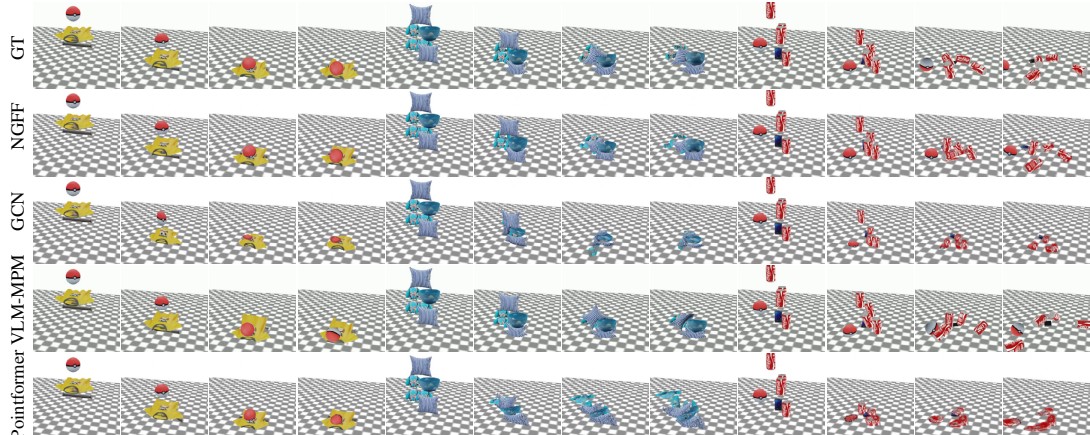

Figure 4: **Qualitative comparison of dynamic prediction methods.** Temporal progression of multi-object scenes demonstrating NGFF's superior trajectory prediction compared to baseline approaches. Each row shows predictions from a different method (NGFF, GCN, Pointformer, Traditional MPM) across identical initial conditions, with time advancing from left to right. The scenarios feature complex, rigid-soft body interactions, including deformable objects (pillows, ropes) interacting with rigid bodies (balls, containers) under gravitational and contact forces. NGFF maintains physically consistent trajectories and realistic deformation patterns throughout extended rollouts, while baseline methods exhibit drift, unrealistic dynamics, or computational instability. Additional dynamic prediction visualizations are provided in Section F.1.

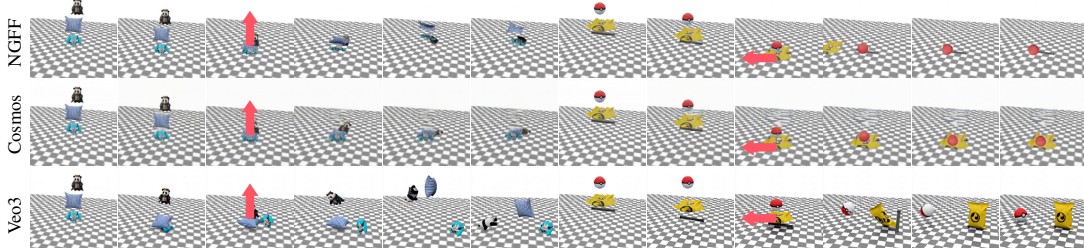

Figure 5: **Interactive generation under external perturbations.** Red arrows indicate applied forces. Left: upward force on fallen pillow; Right: leftward force on cloth affecting ball motion. NGFF produces physically consistent responses to interventions, while baseline methods (Cosmos, Veo3) generate unrealistic dynamics that violate physical constraints. Baseline prompts: Cosmos—"modify the pillow...to show a significant, sudden external force stretching it upward into the air, with interactions with panda and miku"; Veo3—"modify the clothing...to show a significant, sudden external force stretching it leftward."

baseline descriptions and evaluation metrics are detailed in Sections C.1 and C.2. As demonstrated in Table 1 and Figure 4, NGFF consistently outperforms all baselines across generalization dimensions, achieving particularly significant improvements in long-term dynamics and multi-object reasoning scenarios. Notably, our approach delivers approximately two orders of magnitude faster inference

compared to traditional MPM simulators while maintaining superior accuracy, highlighting the computational advantages of learned force field representations.

## 4.3 VIDEO GENERATION

From a video generation perspective, we do not assume ground-truth 3D Gaussians of the initial scenes but acquire them from the RGB images. We evaluate across four key dimensions that capture essential requirements for robust physics-aware video prediction. **Compositional generation** tests adaptation to novel object arrangements and scaling to six-object scenes beyond training complexity. **Novel-view generation** assesses spatiotemporal consistency from unseen camera viewpoints, requiring disentangled dynamics and appearance modeling. **Novel-background generation** evaluates robustness to new visual contexts while preserving physical plausibility. **Interactive generation** probes causal understanding through external perturbations, distinguishing learned physics from trajectory memorization.

Our evaluation employs both automated VLM assessment and human annotation, measuring Physical Realism (PhysR) (physical accuracy) and Photo Realism (PhotoR) (visual quality) as detailed in Section C.2. We compare against leading video generation approaches, including diffusion-based models (NVIDIA Cosmos (NVIDIA et al., 2025), Google Veo3 (DeepMind, 2025)) and physics-engine methods (PhysGen3D (Chen et al., 2025)).

Results in Figure 6 and Table 2 demonstrate that NGFF significantly outperforms existing methods in physical accuracy across unseen scenarios, while achieving competitive visual quality despite modest degradation from 3D reconstruction errors. The object-centric Gaussian representation uniquely enables joint novel-view and novel-background generation capabilities (see Section E). Crucially, Figure 5 illustrates our framework's superior interactive generation: through ODE-based force modeling, NGFF produces physically consistent responses to external interventions, while competing methods fail to maintain plausibility under perturbations.

Table 2: **Video generation performance across generalization scenarios.** Performance metrics (higher is better) evaluated on compositional (Comp.), novel-background (NB), novel-view (NV), and comprehensive (All) splits testing different aspects of generalization capability. The comprehensive split combines all three generalization challenges. Note that Cosmos performs standard novel-view generalization using existing viewpoints, while NGFF tackles the more challenging novel-view synthesis task requiring generation from entirely unseen camera perspectives.

| Model | Split | VLM Eval. | | Human Eval. | |
|---|---|---|---|---|---|
| | | PhysR | PhotoR | PhysR | PhotoR |
| Cosmos | Comp. | 0.34 | **0.42** | 0.29 | 0.43 |
| | NB | 0.26 | **0.46** | 0.30 | 0.41 |
| | NV | 0.39 | 0.42 | 0.26 | 0.39 |
| | All | 0.20 | 0.32 | 0.28 | 0.41 |
| Cosmos tuned | Comp. | 0.26 | 0.35 | **0.57** | **0.58** |
| | NB | 0.38 | 0.36 | 0.60 | 0.60 |
| | NV | **0.49** | **0.40** | **0.63** | **0.62** |
| | All | 0.24 | 0.36 | 0.59 | 0.58 |
| **NGFF-V** | Comp. | **0.47** | **0.42** | 0.56 | 0.55 |
| | NB | **0.56** | 0.42 | **0.63** | **0.61** |
| | NV | 0.44 | 0.38 | 0.55 | 0.54 |
| | All | **0.30** | 0.35 | 0.55 | 0.55 |
| Veo3 | All | 0.29 | **0.41** | 0.53 | **0.64** |
| PhysGen3D | All | 0.19 | 0.35 | **0.57** | 0.58 |

## 4.4 REAL-WORLD EXPERIMENTS

Real-world deployment presents fundamental challenges due to sim-to-real gaps in both perception accuracy and physical property estimation. We evaluate this transition by applying trained models to real-world scenarios and comparing predictions against observed behaviors.

Figure 7 presents comparative results between NGFF and state-of-the-art video generation models (Veo3, Cosmos, and Cosmos fine-tuned on GSCollision). While competing approaches produce visually appealing outputs, they systematically fail to capture accurate gravitational effects and realistic object interactions in real environments. Fine-tuning Cosmos on our synthetic data leads to overfitting behaviors that reduce real-world reliability. In contrast, NGFF generates trajectories that closely align with observed real-world physics, demonstrating effective transfer of learned force field dynamics across the simulation-reality gap. Additionally, the results also demonstrate that the model can generalize to unseen objects and their compositions.

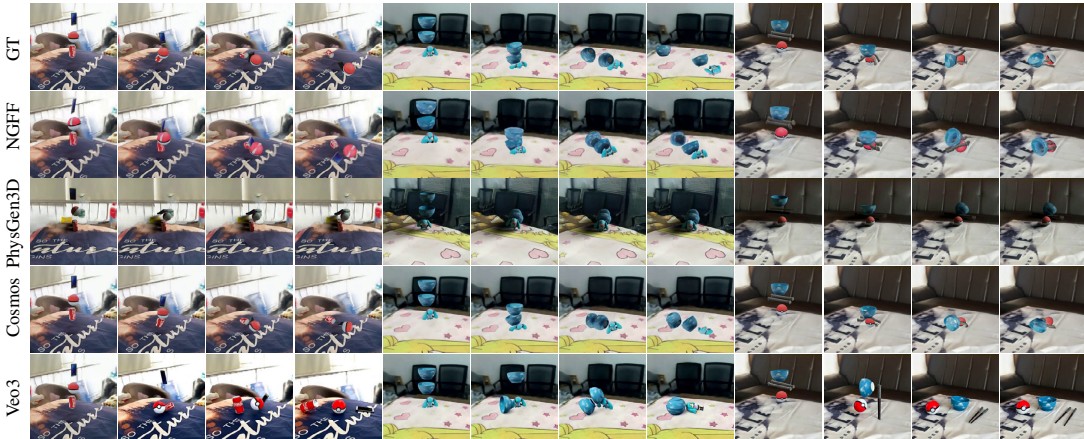

Figure 6: **Video generation quality comparison.** Temporal sequences comparing NGFF against video generation baselines across diverse scenarios. NGFF maintains coherent object shapes, physically plausible interactions, and consistent backgrounds throughout generated sequences, while baseline methods exhibit shape distortions, unrealistic dynamics, and scene inconsistencies that violate physical constraints.

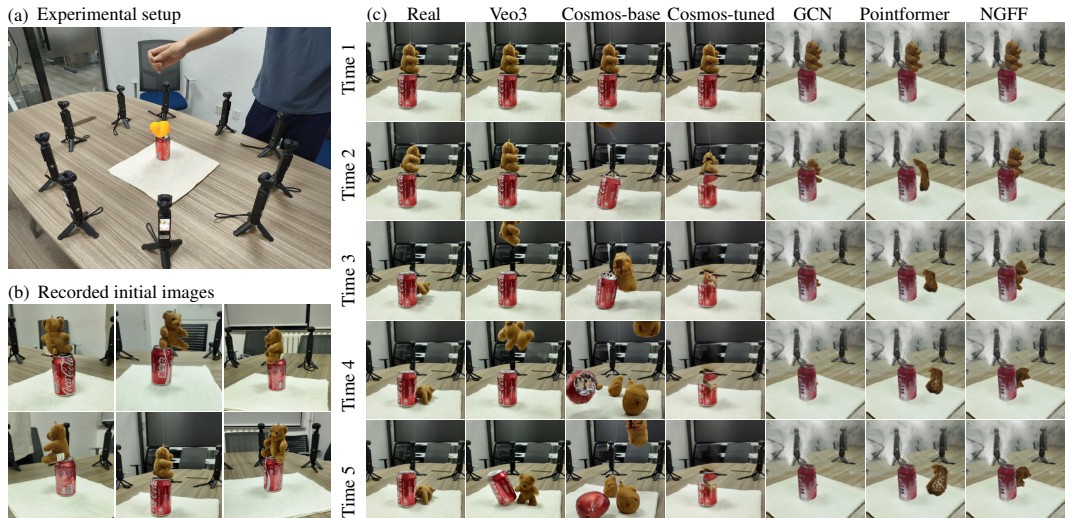

Figure 7: **Real-world validation.** (a) Multi-view capture setup using 10 Pocket 3 cameras recording object dropping experiments. (b) Initial multi-view frames from real-world scenes. (c) Model comparison against ground truth. While video generation models produce visually appealing results, they exhibit physical inconsistencies including object hallucination, unrealistic gravity, and incorrect collision dynamics. NGFF demonstrates superior physical accuracy and consistency with real-world dynamics.

This validation highlights the importance of physics-grounded representations in bridging perception and simulation for real-world applications. Detailed experimental protocols are provided in Section C.3.

## 5 DISCUSSION

**Predicting 4D physical dynamics from minimal observations** While NGFF currently requires multi-view inputs for reliable 3D Gaussian reconstruction, human physical reasoning often operates from single observations or partial views. This observation-reconstruction gap represents a fundamental challenge for deploying physics-aware models in resource-constrained or dynamic environments. Addressing this limitation requires integrating stronger geometric and physical priors into the reconstruction pipeline, potentially through large-scale pretraining on diverse visual-physical datasets or by incorporating generative models that can hallucinate plausible 3D structure from minimal visual evidence. Such advances would bring computational models closer to human-level efficiency in physical scene understanding.

**Scaling to diverse objects and complex scenes**    Our current evaluation spans 10 representative objects across rigid and soft material categories, providing a controlled testbed for physics-grounded reasoning. However, real-world applications demand handling thousands of object types with varying material properties, articulated structures, and complex environmental interactions. Scaling NGFF to this diversity requires fundamental advances in both data efficiency—learning physical principles from limited examples—and representation learning that can generalize across material classes and geometric configurations. Additionally, extending beyond laboratory settings to outdoor environments with complex lighting, weather, and surface conditions presents additional challenges for robust perception-to-physics mapping.

**Interpretable physics-grounded reasoning**    A key advantage of NGFF over black-box video generation approaches lies in its explicit intermediate representations—3D geometries, force fields, and object-centric states—that provide interpretable windows into the model's reasoning process. Future developments could leverage this interpretability for richer forms of physical understanding, including causal counterfactual reasoning that answers "what if" questions through systematic interventions, explicit disentanglement of latent physical properties such as mass and material stiffness, and compositional reasoning about novel physical scenarios. These capabilities would significantly enhance the utility of physics-aware models in scientific discovery, robotics control, and embodied AI applications where understanding causality is as important as prediction accuracy.

**The trade-off between visual quality and physics-grounding**    Our current focus on physical interactions over complex rendering highlights an inherent trade-off between visual fidelity and physical consistency. While 2D video generation methods excel at photorealism—inherently capturing complex lighting and shading effects—they often lack accurate physical grounding. Conversely, 3D-based prediction models ensure explicit physical dynamics but are currently limited by reconstruction artifacts and simplified rendering. We believe that synergizing physics-grounded prediction frameworks with recent advancements in generative reconstruction and video generation models offers a promising path to mitigate these artifacts and significantly elevate visual realism in NGFF.

## 6    CONCLUSION

We introduce NGFF, a unified neural framework that bridges 3D perception and physics simulation through explicit force field modeling over Gaussian representations. By combining feed-forward 3D reconstruction with learned dynamics, it generates physically consistent and visually realistic 4D videos while enabling interactive and novel view synthesis from multi-view RGB inputs.

Comprehensive evaluation demonstrates that NGFF achieves superior performance compared to state-of-the-art video generation and physics simulation methods across spatial, temporal, and compositional generalization scenarios. The framework's object-centric representation and physics-grounded force fields enable robust transfer from synthetic training to real-world scenarios, while maintaining computational efficiency through differentiable Gaussian splatting.

Looking forward, extending NGFF to handle minimal observation requirements, diverse object categories, and complex environmental conditions represents promising directions for developing general-purpose world models. Such advances could enable integrated systems that combine physical consistency with visual realism for robust prediction, causal reasoning, and interactive planning in embodied AI applications. The explicit interpretability of our force field representations provides a foundation for future research in causal physical reasoning and scientific discovery applications.

**Reproducibility statement**    To facilitate reproducibility, we document the data generation process in Section A, provide implementation details of our model in Section B, describe the setup of baseline methods in Section C.1, outline the evaluation metrics in Section C.2, and detail the collection and processing of real-world data in Section C.3. Both the data and code will be released.

ACKNOWLEDGMENTS

This work is supported in part by the Brain Science and Brain-like Intelligence Technology - National Science and Technology Major Project (2025ZD0219400), the National Natural Science Foundation of China (62376009), the State Key Lab of General AI at Peking University, the PKU-BingJi Joint Laboratory for Artificial Intelligence, the Wuhan Major Scientific and Technological Special Program (2025060902020304), the Hubei Embodied Intelligence Foundation Model Research and Development Program, and the National Comprehensive Experimental Base for Governance of Intelligent Society, Wuhan East Lake High-Tech Development Zone.

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

# A    DATASET CONFIGURATION

To support large-scale evaluation of physically grounded video prediction, we construct GSCollision, a dataset that couples Gaussian splatting with MPM-based simulation. The pipeline described in Section A.1 generates temporally consistent Gaussian trajectories from multi-object scenes under controlled physics. The dataset is organized into a modular structure (Section A.2) that includes backgrounds, object assets, scene configurations, simulated dynamics, and video recordings, providing a unified platform for reconstruction and prediction. Finally, we define systematic generalization splits (Section A.3) that cover spatial, temporal, and compositional variations, enabling rigorous testing of model robustness across diverse physical scenarios.

## A.1    DATA GENERATION

To construct our dataset, we employ a hybrid pipeline that integrates Gaussian splatting with a GPU-accelerated MPM engine implemented with Warp (Xie et al., 2024). First, pretrained Gaussian scene representations are loaded from checkpoints and pre-processed by removing low-opacity kernels, applying global rotations, and transforming particles to a normalized coordinate system. If available, segmentation masks are used to reorder particles by object identity, enabling per-object material assignments and stiffness parameters. Optionally, internal particle filling is performed to increase density for more accurate simulation. We utilized Neo-Hookean elasticity as the constitutive model. The number of particles for each object is shown in Table A1.

| | ball | bowl | can | cloth | duck | miku | panda | phone | pillow | rope |
|---|---|---|---|---|---|---|---|---|---|---|
| **Particles** | 106115 | 232764 | 70569 | 116188 | 80544 | 48316 | 56104 | 26571 | 64229 | 8051 |

Table A1: **Objects and their particle counts.**

The pre-processed particle states are then converted into initial conditions for the MPM solver, where particle volumes, covariance matrices, and object-specific material parameters (*e.g.*, Young's modulus, density, boundary conditions) are configured. The simulation domain is defined as a cubic grid. The solver is initialized with either zero or user-specified velocities, and a box-shaped boundary of size 2 is enforced as defined by the scene configuration. Each particle stores position, velocity, deformation gradient, rotation, covariance, stress, mass, and density, which are dynamically updated at each step through the standard particle–grid–particle (P2G2P) pipeline. We simulate each scene for 100 main steps, corresponding to a total of 20,000 substeps.

During simulation, the solver advances dynamics through substeps, exporting per-frame particle attributes including positions, covariance matrices, and rotations. These outputs are saved as frame-wise datasets (*e.g.*, in .h5 format), which preserve all Gaussian attributes required for differentiable rendering. This process produces temporally consistent particle trajectories aligned with Gaussian splatting, yielding high-fidelity dynamic sequences that couple perception and physics.

**How collision are handled?**    In our system, collisions are handled entirely at the grid stage following standard practice in modern MPM frameworks. The simulation proceeds in a P2G to Grid Update to Collision (Grid BC) to G2P pipeline. Collision detection and response are applied after the grid momentum update and before transferring velocities back to particles.

Each collider (plane, cuboid, bounding box, etc.) registers a boundary-condition kernel. During each step, the simulator iterates through these kernels and evaluates whether each grid node lies inside the collider region. When a grid node is detected to be inside (or crossing) a collider, we apply a velocity projection consistent with the collider's physical model. These boundary conditions are imposed directly on the grid velocity field. After enforcing the grid boundary conditions, particles gather updated grid velocities during the G2P stage. Because particle velocities are derived from these corrected grid velocities, particles naturally satisfy non-penetration and frictional constraints without explicit particle-level collision detection.

## A.2    DATA STRUCTURE

GSCollision is organized into several components that together provide a complete pipeline from scene configuration to dynamic simulation:

- **backgrounds** stores environment-specific backgrounds (*e.g.*, `table0`, `table1`). Each subdirectory contains camera parameters (`camera_2999.pt`) and Gaussian point cloud representations (`gaussians_feedforward.ply`), enabling consistent scene reconstruction and rendering.

- **objects** contains individual object assets. Each object (*e.g.*, `ball`, `pillow`) includes camera calibration data (`cameras.json`) and its corresponding point cloud (`point_cloud`), serving as atomic units for scene composition and physical simulation.

- **scene_configs** provides scene-level configuration files (*e.g.*, `3_0.json`, `3_1.json`) that specify object layouts and initialization conditions for simulation.

- **scenes** contains multi-object scene Gaussians grouped by index (*e.g.*, `3_0`, `3_1`). Each scene contains different object combinations (*e.g.*, `0_panda_ball_can`, `300_miku_miku_pillow`), representing diverse interaction setups.

- **mpm** stores dynamic Gaussian trajectories simulated with the *MPM*. Subdirectories mirror those in **scenes**, allowing direct correspondence between scene definitions and their physically grounded dynamics.

- **initial** contains the multi-view images of the initial scene prior to interaction, serving as the starting point for temporal evolution.

- **dynamic** records the dynamic videos of object interactions, aligned with **initial** and **mpm**, and used for training and evaluating video prediction models.

In summary, GSCollision integrates backgrounds, object assets, scene configurations, and both simulated (mpm) and recorded (initial, dynamic) trajectories. This structure enables systematic construction of complex multi-object environments and provides a unified platform for studying **scene reconstruction, physical simulation, and dynamic prediction**.

```
+-- backgrounds
|   +-- table0
|   |   +-- camera_2999.pt
|   |   \-- gaussians_feedforward.ply
|   +-- table1
|       +-- camera_2999.pt
|       \-- gaussians_feedforward.ply
+-- objects
|   +-- ball
|   |   +-- cameras.json
|   |   \-- point_cloud
|   \-- pillow
|       +-- cameras.json
|       \-- point_cloud
+-- scene_configs
|   +-- 3_0.json
|   \-- 3_1.json
\-- scenes
|   +-- 3_0
|   |   +-- 0_panda_ball_can
|   |   \-- 100_can_panda_phone
|   \-- 3_1
|       +-- 300_miku_miku_pillow
|       \-- 301_cloth_can_panda
+-- mpm
|   +-- 3_0
|   |   +-- 0_panda_ball_can
|   |   \-- 100_can_panda_phone
|   \-- 3_1
|       +-- 300_miku_miku_pillow
|       \-- 301_cloth_can_panda
+-- initial
|   +-- 3_0
|   |   +-- 0_panda_ball_can
|   |   \-- 100_can_panda_phone
|   \-- 3_1
|       +-- 300_miku_miku_pillow
```

```
|       \-- 301_cloth_can_panda
+-- dynamic
|   +-- 3_0
|   |   +-- 0_panda_ball_can
|   |   \-- 100_can_panda_phone
|   \-- 3_1
|       +-- 300_miku_miku_pillow
|       \-- 301_cloth_can_panda
```

The directory sizes of the dataset is shown in Table A2.

Table A2: **Directory sizes of the GSCollision dataset.** Others contain objects, config files, reconstruction files, etc.

| Directory | Size (T) | Percentage |
|---|---|---|
| dynamic | 0.854 | 20.1% |
| initial | 0.122 | 2.9% |
| mpm | 2.300 | 54.1% |
| backgrounds | 0.032 | 0.8% |
| scenes | 0.061 | 1.4% |
| others | 0.881 | 20.7% |
| **Total** | **4.250** | **100%** |

## A.3 GENERALIZATION SPLITS

We partition the dataset into 12 groups. Among them, groups 3_0–3_8 serve as the training set, while group 3_9, 4 and 6 are used to test generalization. Table A3 summarizes the dataset configuration and evaluation splits for both dynamic prediction and video generation. The training set is built from object triplets drawn from ten categories, across groups 3_0–3_8, with trajectories spanning 80 simulation steps (1.6s), rendered from 20 viewpoints and 4 backgrounds. For dynamic prediction, we consider three generalization settings: **spatial** (novel object placements in group 3_9), **temporal** (longer rollouts of 100 steps), and **compositional** (novel object combinations involving 4–6 objects in groups 4 and 6). For video generation, we further define splits for **compositional** (3_9, 4, 6), **novel-view** (5 unseen viewpoints), **novel-background** (held-out backgrounds), and a **comprehensive** split that jointly evaluates multiple factors with trajectories extended to 100 steps (2s). Green cells indicate aspects consistent with training, while blue cells denote novel conditions used for testing.

Table A3: **Statistics of different generalization splits.** Green indicates that the training and test data share the same configuration in certain aspects, whereas blue indicates they are different.

| | Objects | Groups | Time span | Viewpoints | Backgrounds |
|---|---|---|---|---|---|
| Training set | 3 from 10 kinds | 3_0 − 3_8 | 80 step / 1.6s | 20 | 4 |
| **Dynamic prediction** | | | | | |
| Spatial | 3 from 10 kinds | 3_9 | 80 step / 1.6s | / | / |
| Temporal | 3 from 10 kinds | 3_0 − 3_8 | 100 steps / 2s | / | / |
| Compositional | 4–6 from 10 kinds | 4, 6 | 80 step / 1.6s | / | / |
| **Video generation** | | | | | |
| Compositional | 3–6 from 10 kinds | 3_9, 4, 6 | 80 step / 1.6s | 20 | 4 |
| Novel-view | 3 from 10 kinds | 3_0 − 3_8 | 80 step / 1.6s | 5 | 4 |
| Novel-background | 3 from 10 kinds | 3_0 − 3_8 | 80 step / 1.6s | 20 | 4 |
| Comprehensive | 3–6 from 10 kinds | 3_9, 4, 6 | 100 steps / 2s | 5 | 4 |

# B IMPLEMENTATION DETAILS

## B.1 FEED-FORWARD GAUSSIAN RECONSTRUCTION

Starting from uncalibrated RGB images, the initial step is to recover the 3D point cloud structure of a scene. Traditional optimization-based methods, such as Structure-from-Motion and Multi-View Stereo, necessitate capturing tens or even hundreds of views, which are often impractical in real-world scenarios. Recently, feed-forward foundation reconstruction models (Wang et al., 2025a;b) have emerged as a powerful alternative. Pretrained on massive datasets, these models perform 3D reconstruction in a single forward pass, enabling lightning-speed scene reconstruction. This provides the foundation for subsequent neural simulation and planning within the reconstructed 3D representations.

In our experiments, we found that the permutation-equivariant architecture of $\pi^3$ achieves higher accuracy in object registration compared to VGGT, a model based on first reference frame reconstruction. Consequently, we selected $\pi^3$ as our backbone.

Building upon the $\pi^3$ model, we introduce modifications to create $\pi^3 - GS$ for feed-forward Gaussian scene reconstruction. To achieve stronger real-world generalization, we freeze the alternating attention encoder and the camera head of the pre-trained $\pi^3$ model. We directly use the predictions from its point head as the centers, $\mu$, for the Gaussians. Furthermore, we observed that MLP-based pixel-shuffling is prone to creating artifacts at patch boundaries. Since convolutional operations yield smoother results, we replaced this with a convolutional upsampling layer in the splatter head. Specifically, we first refine the patch features from the transformer encoder with three convolutional blocks, followed by an upsampling layer and two additional convolutional blocks to eliminate artifacts. We also applied a direct RGB shortcut (Ye et al., 2025), composed of 3 Residual CNN blocks from the input image, to preserve high-frequency details and enhance appearance reconstruction.

We trained the splatter head of our $\pi^3 - GS$ model on the Wildrgbd (Xia et al., 2024) dataset, which contains approximately 22,000 scenes. The training was conducted on 8 NVIDIA H100 80G GPUs for 50 epochs with a global batch size of 24. Both mixed-precision training and gradient checkpointing were utilized.

## B.2 SINGLE-VIEW GAUSSIAN REFINEMENT

Feed-forward reconstruction models that lack a generative prior are inherently limited in handling challenges such as incomplete observations and occlusions. This deficiency can adversely affect the topological integrity of the object's 3D Gaussian representation and, consequently, the fidelity of subsequent neural simulations. To address this, we propose a pipeline that first completes the object's geometry using a 3D asset generation model, followed by a $\mathrm{Sim}(3)$ point cloud alignment to register it within the scene.

Initially, the segmented object image is processed through a super-resolution pipeline (Wu et al., 2024a) to enhance textural details. We then employ a pretrained 3D generative model, DiffSplat (Lin et al., 2025a), to infer a complete 3D Gaussian representation of the object, conditioned on the single input view.

The generated Gaussian asset resides in a normalized, object-centric coordinate system, which is inconsistent with the object's true scale and pose in the scene. To place the generated object accurately, we introduce a $\mathrm{Sim}(3)$ registration algorithm that combines visual feature matching with gradient-based optimization. First, we render a set of images $\{\mathcal{I}_k\}$ by orbiting the generated asset at multiple elevations. For each rendered image $\mathcal{I}_k$, we use SuperGlue (Sarlin et al., 2020) to establish matches with the original input image $\mathcal{I}_{in}$, and select the view that yields the maximum number of 2D correspondences, denoted as $\mathcal{C}_{2D} = \{(\mathbf{p}_i, \mathbf{p}'_i)\}_{i=1}^N$. These 2D matches are then lifted to 3D, $\mathcal{C}_{3D} = \{(\mathbf{P}_i, \mathbf{P}'_i)\}_{i=1}^N$, by identifying the 3D points in the respective point clouds, $\mathcal{P}_{gen}$ and $\mathcal{P}_{obs}$, that are closest to the corresponding camera rays. For initialization, we estimate the scale $s_{init}$ from the ratio of the point clouds' bounding box volumes and solve for an initial 6-DoF pose $[\mathbf{R}_{init}|\mathbf{t}_{init}] \in SE(3)$ using the Kabsch algorithm within a RANSAC framework. Subsequently, we jointly refine the similarity transformation $\mathbf{T} \in \mathrm{Sim}(3)$ by minimizing the Chamfer distance

between the transformed generated point cloud and the observed point cloud via gradient descent:

$$(\mathbf{R}^*, \mathbf{t}^*, s^*) = \arg \min_{\mathbf{R} \in SO(3), \mathbf{t} \in \mathbb{R}^3, s \in \mathbb{R}^+} \mathcal{L}_{CD}(s\mathbf{R}\mathcal{P}_{gen} + \mathbf{t}, \mathcal{P}_{obs})$$

The entire registration process can be done within a few seconds.

### B.3 NEURAL GAUSSIAN FORCE FIELD (NGFF)

Our framework builds upon a neural interaction–based dynamics predictor, which integrates object-level interaction modeling, boundary constraints, and stress field prediction into a differentiable ODE solver. The overall design couples four components: an Interaction Network (IN), a Stress Prediction Network (StressNet), boundary and collision modules, and a neural ODE–based temporal evolution module.

**Interaction Network (IN)**  The IN module captures both geometric and state-dependent interactions among multiple objects. Each object is first encoded using a hierarchical PointNet backbone that extracts global geometric features from point clouds. Center of mass (CoM), orientation angles, linear and angular velocities are embedded through multilayer perceptrons. Pairwise object relations are modeled via a branch–trunk structure: branch features encode relative states between objects, while trunk features preserve object-specific information. Their interaction is combined through element-wise multiplication and mapped to output forces and torques. To ensure physical consistency, the IN explicitly detects inter-object collisions and boundary contacts. Collision forces are masked by an intersection matrix, while boundary forces are predicted by a dedicated boundary network conditioned on both geometry and state features.

**Stress prediction (StressNet)**  Beyond rigid-body dynamics, the model accounts for distributed internal responses by predicting per-point stress fields. StressNet takes as input the local point coordinates, velocities, and the aggregated forces and torques from IN and boundary interactions. A shared MLP extracts local features, followed by a global max-pooling to capture object-level context. These are fused and projected to pointwise stress outputs. The design enforces rotation consistency by transforming predicted forces and stresses between global and local frames via differentiable Euler-angle rotation matrices.

**Boundary and collision modules**  Physical validity is further maintained through two auxiliary functions: collision detection computes pairwise point distances between objects to construct overlap masks, which gate non-contact interactions; boundary detection evaluates the proximity of object points to the simulation domain limits, producing boundary masks to trigger repulsive boundary forces.

**Temporal evolution with neural ODE**  To simulate motion, NGFFobj integrates the above predictors into a continuous-time dynamics system solved via the torchdiffeq ODE framework. The system state comprises point positions, point velocities, CoM and angular states, along with stress distributions. At each step, the IN outputs interaction forces and torques, and StressNet provides stress derivatives, which are combined with external forces (if any) and gravity. The resulting accelerations are integrated forward in time using either explicit Euler or adaptive-step solvers. This formulation enables stable long-horizon rollout while preserving differentiability for learning-based optimization.

**Training**  The model is trained on 8 NVIDIA H100 80GB GPUs for 1001 epochs for 48 hours. The learning rate starts at $1 \times 10^{-5}$ and decays to a minimum of $1 \times 10^{-7}$. The architecture consists of 4 layers with a hidden dimension of 200. The batch size is set to 9 per node, and each epoch involves 80 steps with a chunk size of 80. The ODE method used is Euler with a step size of $2 \times 10^{-2}$, and a threshold of $5 \times 10^{-2}$ for collision detection is applied during training.

## C   EXPERIMENTAL SETUP

### C.1   BASELINES

#### C.1.1   GRAPH CONVOLUTIONAL NEURAL NETWORKS (GCN)

We adopt a Graph Convolutional Network (GCN) to model dynamics. Each object is represented by a set of keypoints, which serve as graph nodes, and edges are constructed using a radius-based neighbor search with a threshold. The node features are obtained by concatenating the 3D position and velocity of each keypoint.

The network consists of multiple GCNConv layers, where each layer performs message passing to aggregate information from neighboring nodes, followed by ReLU nonlinearities. A final fully connected layer predicts the residual update of position and velocity for each node. Prediction is performed in an autoregressive manner: at each step, the model updates the current state with the predicted residuals and rolls out the trajectory over multiple steps.

The GCN is trained on a single NVIDIA H100 80GB GPU with a learning rate starting at $1 \times 10^{-3}$, which decays to $1 \times 10^{-4}$. The model consists of 4 layers, each with a hidden dimension of 128. A batch size of 30 is used, with 80 steps per epoch, and the training runs for 500 epochs. At each step, the model processes 3000 samples, with data processed in chunks of 80 to ensure efficient memory usage. The dynamic model used in this setup is GCN, which is specifically designed to handle graph-structured data and learn complex relationships.

#### C.1.2   POINTFORMER

Pointformer directly models interactions across all object keypoints. Each keypoint is embedded using a positional encoding derived from its 3D coordinates, followed by a linear projection into a high-dimensional latent space. The set of embedded keypoints from all objects is then processed by a stack of multi-head self-attention layers, allowing each point to attend to and aggregate information from all others in the scene.

To handle variable numbers of objects and keypoints, a padding mask is applied to prevent attention from propagating through invalid nodes. The transformer output is normalized and projected back into the point space via a feedforward head to predict residual updates for each keypoint's position. As in the GCN baseline, prediction proceeds autoregressively over multiple rollout steps, generating a sequence of future trajectories.

Unlike GCNs, which rely on local neighborhood graphs, PointFormer captures global interactions across all keypoints through self-attention. This enables the model to represent long-range dependencies and complex multi-object dynamics, but at the cost of higher computational complexity due to quadratic attention scaling.

The Pointformer is trained on 4 NVIDIA H100 80GB GPUs for 60 hours. The model is trained with a learning rate starting at $5 \times 10^{-4}$, decaying to a minimum of $5 \times 10^{-6}$. The architecture consists of 3 layers and a hidden dimension of 128, with dropout 0.1. The batch size is set to 8 per node, with a total of 2001 epochs, and each epoch involves 80 steps with a chunk size of 80.

#### C.1.3   VLM-MPM

We employ Gemini-2.5-flash to infer the Young's modulus and density from 20 training videos. The estimated parameters are subsequently normalized to align with the value ranges required by the MPM simulator. The prompt used is:

```
For each object in the videos, estimate the object's density in kilograms
    per cubic meter and its Young's modulus in Pa. Return an json array
    of objects in JSON where each object has fields: name, density,
    youngs_modulus. Do not include extra text, only valid JSON that
    matches the schema. The objects you need to estimate are: {objects}.
```

The following simulations are identical to those employed in data generation.

### C.1.4 COSMOS-PREDICT2

Cosmos-Predict2 is a World Foundation Model trained by NVIDIA, designed to simulate and predict the future state of the world as video. It can serve as a foundation for training physical AI systems in digital environments. The model balances both visual quality and physics awareness and is capable of generalizing to downstream tasks with a small amount of post-training.

We performed full-parameter fine-tuning on the **Cosmos-Predict2-2B-Video2World-480P-16FPS** model. For this process, we utilized a total of 216K video clips from the GSCollision dataset, which amounts to 17.28 million frames, each with a resolution of $448 \times 448$.

For text conditioning, we used the following prompt for all video clips:

```
A photorealistic video. Simulate the future dynamics of the foreground
    objects falling from the air onto the table. The simulation should
    realistically model various physical interactions, including
    deformation, gravity, collisions between the objects, and their
    impact with the surface. Capture the subsequent motions until the
    objects come to a complete rest.
```

For image (video) conditioning, we randomly used 3-5 latent frames (corresponding to 9-17 actual frames) during training. During testing, we conditioned on the first 13 frames of the video.

The training was conducted on 8 NVIDIA H100 80G GPUs. We trained for 20,000 iterations until convergence, using an initial learning rate of $2.5 \times 10^{-4}$ and a global batch size of 24.

### C.1.5 PHYSGEN3D

PhysGen3D transforms a single static image into an interactive, amodal 3D scene capable of simulating physically plausible future outcomes. The framework first reconstructs a complete 3D world by leveraging a suite of pretrained vision models to infer geometry, semantics, materials, and lighting properties from the input image. This reconstructed scene is then passed to a physics-based simulator, which uses the MPM to generate object dynamics in response to LLM-inferred physical parameters. Finally, a physics-based rendering module seamlessly composites the simulated dynamic objects and their corresponding shadows back into the original scene, producing a coherent and controllable video. PhysGen3D enables fine-grained control over object interactions and generates motions that adhere to physical laws.

However, the framework's reliance on single-view reconstruction makes it susceptible to errors in complex scenarios. The method is primarily designed for scenes with simple geometry and can fail when dealing with heavy occlusions and multiple objects. The ill-posed nature of inferring 3D properties from a 2D image can lead to perception failures and parameter estimation errors under challenging situations. Besides, reliance on MPM simulators makes it slower than neural simulation methods on modern GPUs.

### C.1.6 VEO3

Veo3 is a SOTA diffusion-based video generation model developed by Google DeepMind. It can interpret complex text prompts, capable of generating smooth and consistent dynamics for people and objects. It avoids the uncanny or jarring artifacts common in earlier models, producing motion that is both believable and visually pleasing.

However, during testing, we observed that while Veo3 maintains excellent temporal consistency during non-strenuous motion, the model still frequently generates outputs that violate fundamental Newtonian physics principles or object permanence during strenuous events, such as collisions.

### C.2 EVALUATION METRICS

In this study, we adopt different metrics for evaluating NGFF. For the accuracy of the predicted dynamics, we choose RMSE, FPE, and R as our primary metrics. For assessing the video generation correctness, we employ PhysR and PhotoR.

**Root Mean Squared Error (RMSE)**   The RMSE, defined as the square root of the MSE, retains the property of penalizing larger deviations but expresses the error in the same units as the original data. This makes it easier to interpret in physical contexts, as it reflects the average magnitude of prediction errors relative to true trajectories:

$$\text{RMSE} = \sqrt{\frac{1}{n} \sum_{t=1}^{n} (\hat{z}_t - z_t)^2}. \tag{A1}$$

**Final Position Error (FPE)**   The FPE evaluates the difference between the predicted and ground-truth final positions of an object. This metric is particularly important for goal-oriented physical reasoning, where accuracy at the endpoint is critical. By focusing on the final state, FPE complements trajectory-based metrics and ensures that models not only capture motion dynamics but also predict the ultimate destination correctly:

$$\text{FPE} = |\hat{z}\text{final} - z\text{final}|. \tag{A2}$$

**Position Change Error (PCE)**   The PCE measures the discrepancy between the predicted and actual changes in position over time. This metric can be interpreted as an indicator of how accurately the model captures the object's velocity throughout its motion:

$$\text{PCE} = |\Delta \hat{z}_t - \Delta z_t|. \tag{A3}$$

**Pearson Correlation Coefficient (R)**   The R coefficient captures the linear correlation between predicted and actual trajectories. Rather than measuring absolute error, it reflects how well the model aligns with the overall trajectory pattern. A high value indicates strong agreement in motion trends, while a low value suggests that the model fails to capture the underlying trajectory structure:

$$\text{R} = \frac{\sum_{t=1}^{n} (\hat{z}t - \bar{\hat{z}})(z_t - \bar{z})}{\sqrt{\sum t = 1^n (\hat{z}t - \bar{\hat{z}})^2 \sum t = 1^n (z_t - \bar{z})^2}}. \tag{A4}$$

Given that video-generation models like Veo3 and PhysGen3D are closed-source or untrainable, for which a direct comparison with ground truth would be inequitable, we adopted the qualitative evaluation framework established by PhysGen3D (Chen et al., 2025) to quantitatively evaluate video generation quality. This involves leveraging a Vision-Language Model, Gemini-2.5-flash to assess two key criteria: PhysR and PhotoR.

**Physical Realism (PhysR)**   The PhysR measures how realistically the video follows the physical rules like collision and gravity and whether the video represents real physical properties like elasticity and friction.

**Photo Realism (PhotoR)**   The PhotoR measures the overall visual quality of the video, including the visual artifacts, discontinuities, and id-inconsistency.

The prompt is as follows:

```
# [video inputs]
I would like you to evaluate the quality of generated videos above based
   on the following criteria: physical realism and photorealism. The
   evaluation will be based on 10 evenly sampled frames from each video.
    Given the original image and the above instructions , please
   evaluate the quality of each video on the two criteria mentioned
   above. Note that: Physical Realism measures how realistically the
   video follows the physical rules and whether the video represents
   real physical properties like elasticity and friction. To discourage
   completely stable video generation, we instruct respondents to
   penalize such cases. Photorealism assesses the overall visual quality
    of the video, including the presence of visual artifacts,
   discontinuities, and how accurately the video replicates details of
   light, shadow, texture, and materials. Please provide the following
   details for each video in an json array of videos where each video
   object has fields: physical_realism score, photorealism score and
   content. The content should be a sentence summarizing the video,
   scores should be ranging from 0-1, with 1 to be the best, round to 2
   decimal places:
```

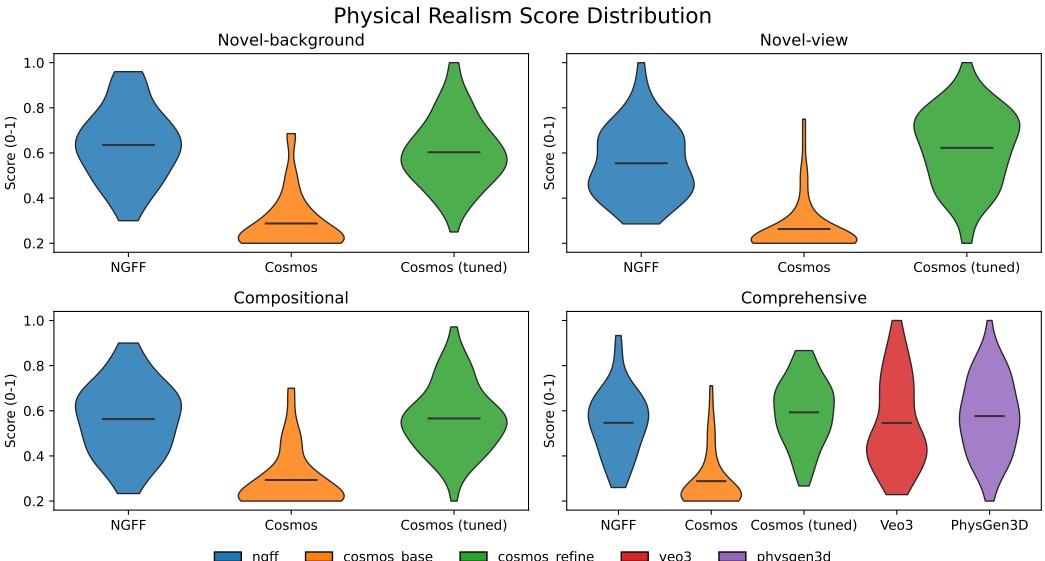

Figure A1: **An example of human study questionaire.**

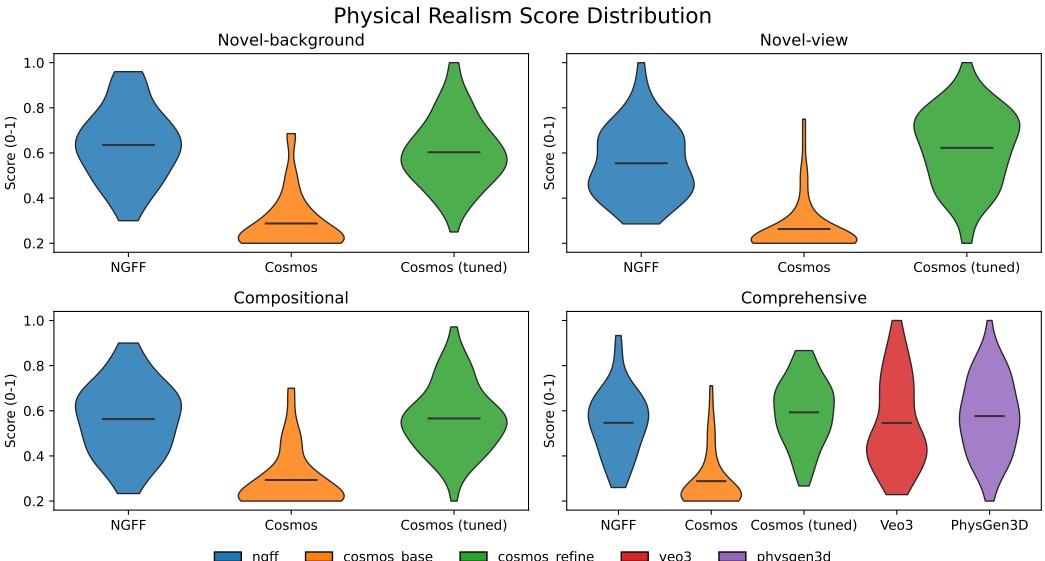

Figure A2: **Detailed distribution of human evaluation results on Physical Realism (PhysR).**

**Human evaluation**   We designed a questionnaire to conduct human evaluation on video generation quality across different models, as illustrated in Figure A1. A total of 61 participants were recruited to complete an 80-page questionnaire. At the beginning, we provided a detailed explanation of two metrics. Each page of the questionnaire contains a 2–3 second video randomly chosen from all models and generalization splits. Participants are instructed to assess each video based on the two dimensions above: PhysR and PhotoR. This human study design, accompanied by results from VLMs, ensures a fair, consistent, and comprehensive evaluation. Detailed distributions of human evaluation results can be found in Figure A2 and Figure A3

## C.3   REAL WORLD ENVIRONMENTS

### C.3.1   DATA COLLECTION

We collected real-world interaction sequences using a multi-view setup of ten DJI Pocket 3 cameras arranged around a table in a standard office environment. All cameras were calibrated to share identical intrinsic parameters, ensuring geometric consistency across views. To induce controlled dynamics, objects were lifted and released with transparent fishing line, creating falling and collision events while guaranteeing that each object started from a static state. In total, we recorded 40 dynamic sequences at 50 FPS and 3K resolution. The object set included a cola can, a teddy bear, and a rubber duck, allowing us to generate diverse two-object and three-object interaction scenarios with varying mass and material properties.

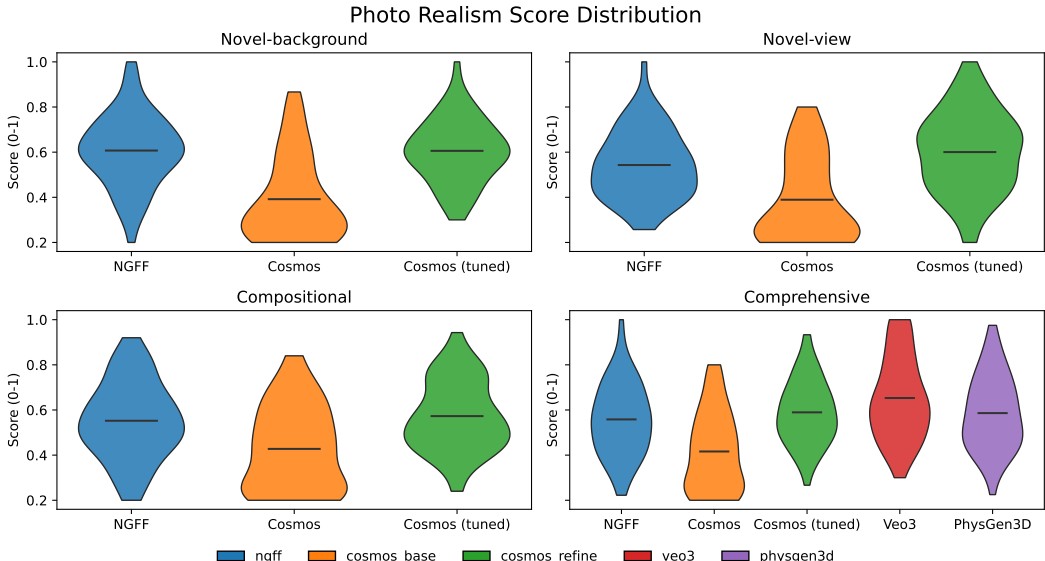

Figure A3: **Detailed distribution of human evaluation results on Photo Realism (PhotoR).**

### C.3.2 VIDEO PROCESSING

For each sequence, we temporally trimmed the videos from the instant of release until all objects came to rest, typically spanning 50–60 frames. Each frame from every camera view was annotated with axis-aligned bounding boxes, obtained semi-automatically using SAM2 and refined by manual correction where necessary to ensure pixel-level accuracy. Object identities were explicitly labeled to support subsequent use in multimodal learning tasks. To enable 3D reconstruction, all frames were synchronized across views and processed using a feed-forward pretrained Gaussian-splatting model, with further refinement using DiffSplat (Lin et al., 2025a), producing multi-view-consistent 3D Gaussian representations. This pipeline ensured both high-quality geometry recovery and consistent object-level alignment, establishing a reliable benchmark for evaluating dynamic prediction models under real-world conditions. See the representative recorded videos in Figure A4, Figure A5, and Figure A6.

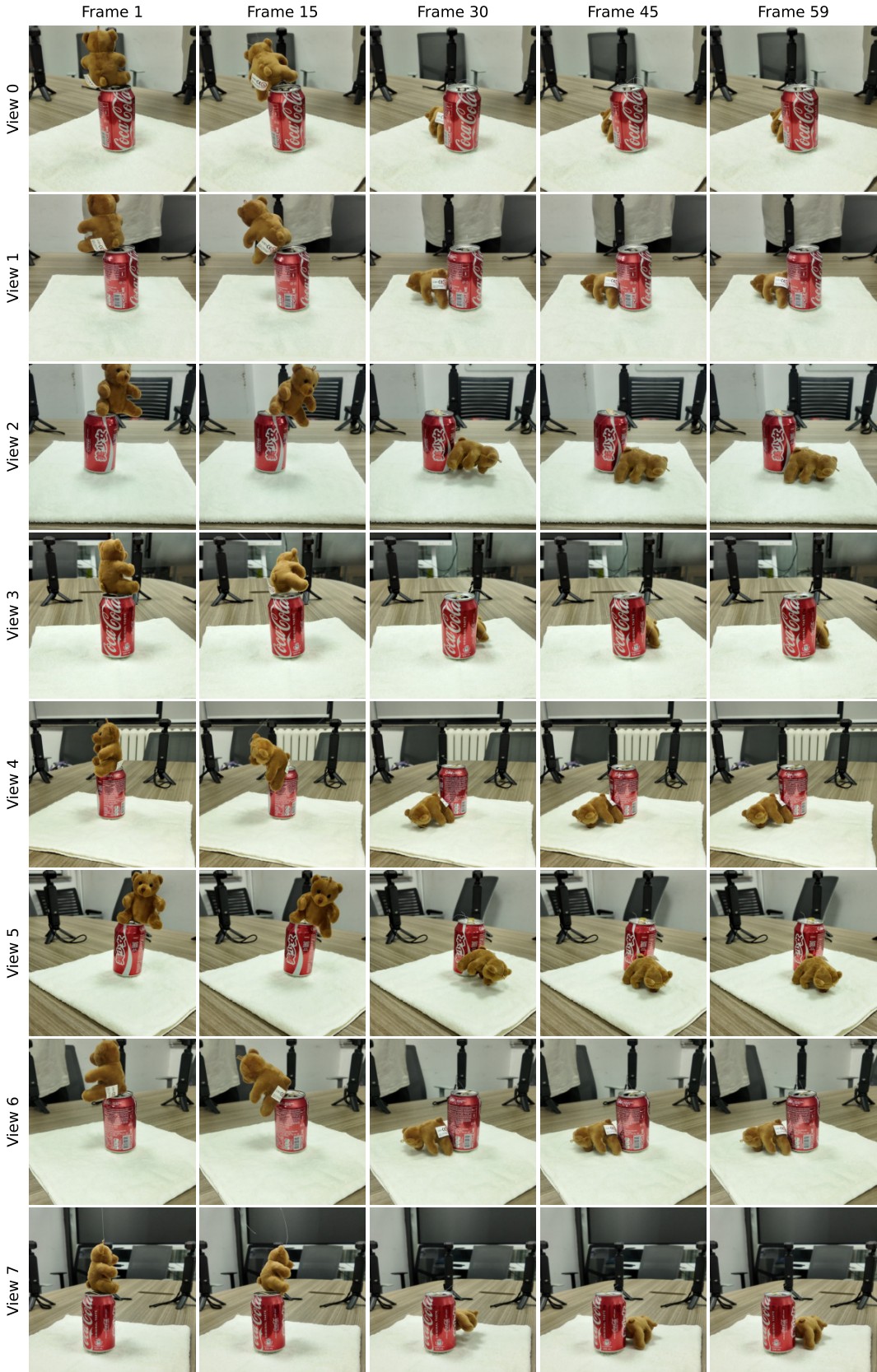

Figure A4: **Recorded multi-view dynamic interaction in the real world.** A teddy bear is released above a cola can and falls onto the table.

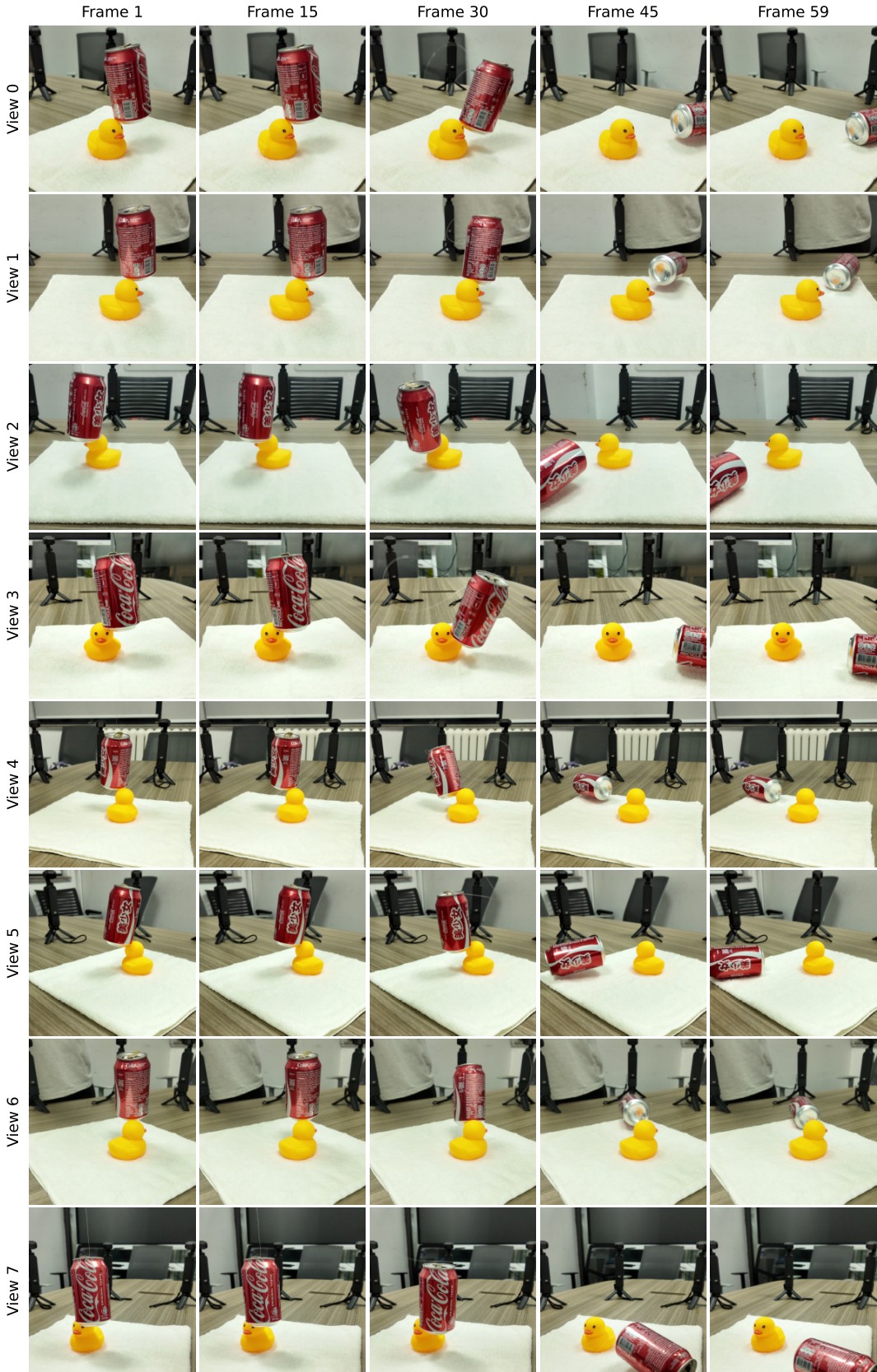

Figure A5: **Recorded multi-view dynamic interaction in the real world.** A cola can is released above a duck and falls onto the table.

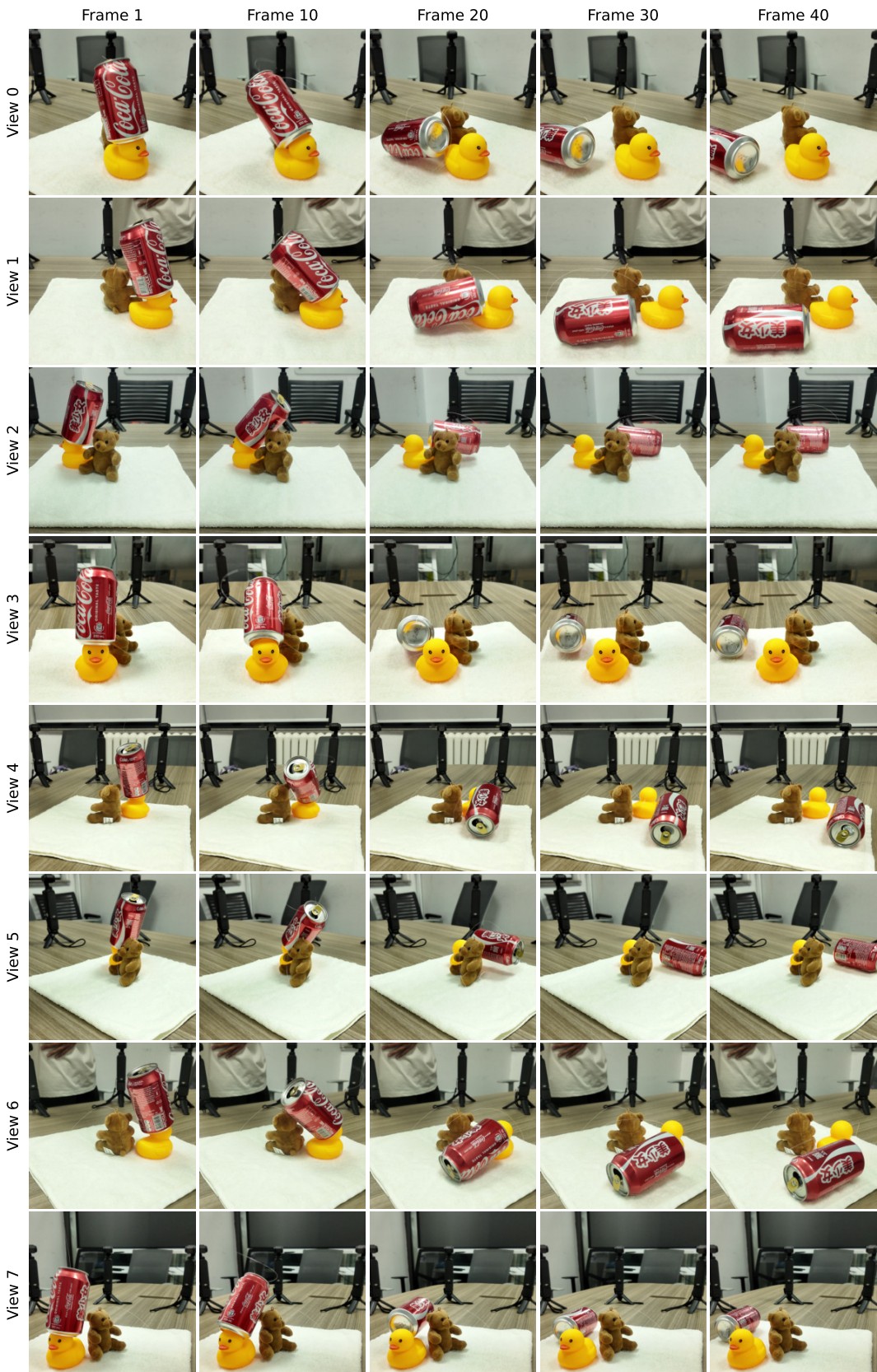

Figure A6: **Recorded multi-view dynamic interaction in the real world.** A cola can is released above a duck, collides with the teddy bear, and falls onto the table.

# D RELATED WORK

## D.1 PHYSICAL REASONING

Physical reasoning is a core human ability to understand and interact with the physical world. Besides generating continuous and high-fidelity videos (Ho et al., 2022; Yang et al., 2024), physical reasoning tackles the challenges to comprehend and reason about the governing physical dynamics of visual scenes, representing a core capability required for AI systems to achieve human-level intuitive physics abilities. This core skill encompasses two critical domains: First, it involves spatial reasoning (Shiri et al., 2024) from video inputs, the ability to reconstruct and understand three-dimensional scenes, including object relationships, spatial configurations, and perspective. Second, it requires an understanding of fundamental physical laws governing object interactions and generalizing it to OOD scenarios.

Various benchmarks have been proposed to assess the physical reasoning capabilities of both humans and machines. Previous studies build datasets based on the VoE paradigm to examine agents' understanding of basic physical concepts (Piloto et al., 2022; Dai et al., 2023). Some studies extend the passive observation paradigm to interactive environments, which require the agents to apply actions to finish tasks (Bakhtin et al., 2019; Allen et al., 2020; Bear et al., 2021; Li et al., 2024). Recent benchmark studies have investigated evaluating the capabilities of VLM (Chow et al., 2025) and video generation models (Bansal et al., 2024; Li et al., 2025a; Duan et al., 2025) in physical world understanding and adherence. These efforts predominantly focus on evaluating the physical commonsense of foundational video generation models in open-world scenarios. The results indicate a significant gap between the performance of current SOTA models and human capabilities (Motamed et al., 2025; Bordes et al., 2025). Our work builds upon the interactive physical environment to demonstrate the reasoning capability of our model.

## D.2 VISUAL DYNAMIC PREDICTION

Visual dynamic prediction, the task of predicting future frames from visual inputs, has been addressed through diverse approaches. Neural simulator-based methods commonly employ GNN as their dynamics backbone due to their relational inductive bias. Early approaches, while capable of simulating various physical phenomena (Sanchez-Gonzalez et al., 2020; Bear et al., 2021), often fail on complex materials and physical interactions. More recent approaches inject physics inductive bias into simulation such as mesh (Allen et al., 2023a) or SDF (Rubanova et al., 2024) representation for rigid bodies and spring-mass models (Jiang et al., 2025a) or particle-grid representations (Zhang et al., 2025) for deformable objects. Despite their advancements, these methods often struggle with complex multi-object interaction scenarios and exhibit limited generalization abilities. While our method adopts a unified representation for different object materials and physical interactions by predicting force fields.

In contrast, physics simulator-based methods explicitly model scene dynamics using differentiable simulators. For example, techniques that render scenes into particles via 3D Gaussian rendering and simulate their evolution with Material Point Method (MPM)-based simulators (Xie et al., 2024; Lin et al., 2025b; Chen et al., 2025) produce realistic outcomes but rely heavily on strong physics priors or case-specific optimization, which may not be available in intuitive physics scenarios.

Diffusion-based video generation models (Blattmann et al., 2023; DeepMind, 2025; NVIDIA et al., 2025) pretrained on massive-scale videos have emerged as powerful generative world models. These models demonstrate a remarkable ability to synthesize temporally coherent and visually compelling sequences of digital frames. However, a fundamental challenge is that they struggle to produce physically coherent frames, lacking an inherent understanding of physical laws (Kang et al., 2025; Motamed et al., 2025). While these models may exhibit plausible dynamic outcomes for simple constrained scenarios, their grasp of physics is superficial, as they tend to retrieve trajectories from the training data rather than adhering to consistent learned physical laws (Kang et al., 2025). Recent studies have explored integrating physical principles into diffusion-based models to generate realistic and controllable object dynamics. Some approaches distill physical priors from video models to infer intrinsic object properties to drive physics simulators (Zhang et al., 2024; Huang et al., 2025). Other works treat physics as an explicit guiding signal or conditional input for the generative process. PhysAnimator (Xie et al., 2025) uses motion sketches from a preliminary physics simulation

Table A4: **Ablation results of NGFF and NGFF without deformation across different generalization settings.** Arrows indicate whether higher (↑) or lower (↓) values are better.

| Setting | Method | MSE (↓) | RMSE (↓) | PCE (↓) | FPE (↓) | PCC (↑) |
|---|---|---|---|---|---|---|
| Spatial | NGFF w/o deform. | 0.01466 | 0.10971 | 0.01386 | 0.45927 | 0.59506 |
| | NGFF | 0.00835 | 0.08199 | 0.01165 | 0.32576 | 0.66111 |
| Temporal | NGFF w/o deform. | 0.02605 | 0.14403 | 0.01421 | 0.59975 | 0.57836 |
| | NGFF | 0.01471 | 0.10711 | 0.01167 | 0.41933 | 0.65238 |
| Compositional-4 | NGFF w/o deform. | 0.02092 | 0.13031 | 0.01487 | 0.54689 | 0.52474 |
| | NGFF | 0.01052 | 0.09533 | 0.01210 | 0.37274 | 0.59444 |
| Compositional-6 | NGFF w/o deform. | 0.01910 | 0.13249 | 0.01527 | 0.54564 | 0.50577 |
| | NGFF | 0.01379 | 0.11268 | 0.01358 | 0.44583 | 0.54707 |

Table A5: **Inference time for different video generation methods** Times are measured on a single NVIDIA H100 80G GPU.

| Model | Time |
|---|---|
| **NGFF-V** | 37s (3 objects) / 72s (6 objects) |
| **NGFF-V (w/o refine)** | 12s (3 objects) / 19s (6 objects) |
| **Pointformer-V** | 37s (3 objects) / 72s (6 objects) |
| **GCN-V** | 37s (3 objects) / 72s (6 objects) |
| **PhysGen3D** | 400s (3 objects) / 590s (6 objects) |
| **Cosmos-predict2-2B** | 20s |
| **Veo3** | 11–360s (via API) |

to guide a video diffusion model, while ForcePrompting (Gillman et al., 2025) introduces forces as a direct control signal, training a model to generate responses to user interactions.

### D.3 SCENE REPRESENTATIONS FOR SIMULATION AND RENDERING

Early methods extracted geometry, such as point clouds, directly from RGB-D inputs for simulation and planning (Shi et al., 2024) and trained a separate module for rendering (Whitney et al., 2024). Later, Neural Radiance Fields (NeRF) enables differentiable rendering and can be jointly optimized for simulation (Driess et al., 2023; Xue et al., 2023a) at the cost of degraded flexibility due to implicit encoders. 3D Gaussian Splatting has emerged as a powerful alternative, offering photorealistic quality and real-time performance (Kerbl et al., 2023). The utility of 3D Gaussians extends beyond static rendering; works like PhysGaussian (Xie et al., 2024) have integrated them with Newtonian dynamics for high-quality motion synthesis. Advances in feed-forward reconstruction significantly accelerate the reconstruction process by directly inferring Gaussian attributes from unposed multi-view images (Wang et al., 2025a; Jiang et al., 2025b; Wang et al., 2025b), enabling fast, end-to-end scene creation suitable for downstream simulations. Our concurrent work 3DGSIM (Zhobro et al., 2025) also employs feed-forward Gausssian reconstruction and a transformer for prediction. They primarily focus on single-object dynamics, having limited generalization to multi-object interactions and planning capability.

## E ABLATIONS AND ADDITIONAL RESULTS

In this section, we provide supplementary results to further analyze the effectiveness and efficiency of our proposed framework. First, we report an ablation study in Table A4, which compares NGFF against its variant without deformation modeling across different generalization settings. The results demonstrate that explicitly modeling deformation consistently improves predictive accuracy, yielding lower errors (MSE, RMSE, PCE, and FPE) and higher correlations (PCC).

We also benchmark the inference speed of our method against alternative approaches in Table A5. NGFF-V attains efficient inference (2.5s per sequence on a single H100 GPU), significantly outperforming computationally expensive physics-based simulators (*e.g.*, PhysGen3D) while remaining competitive with large-scale generative models (*e.g.*, Cosmos-predict2-2B and Veo3). Together,

these results highlight that our framework achieves a favorable balance between accuracy, realism, and efficiency.

# F ADDITIONAL VISUALIZATIONS

## F.1 DYNAMIC PREDICTION

We present more visualizations of dynamic prediction in Figures A7 to A11.

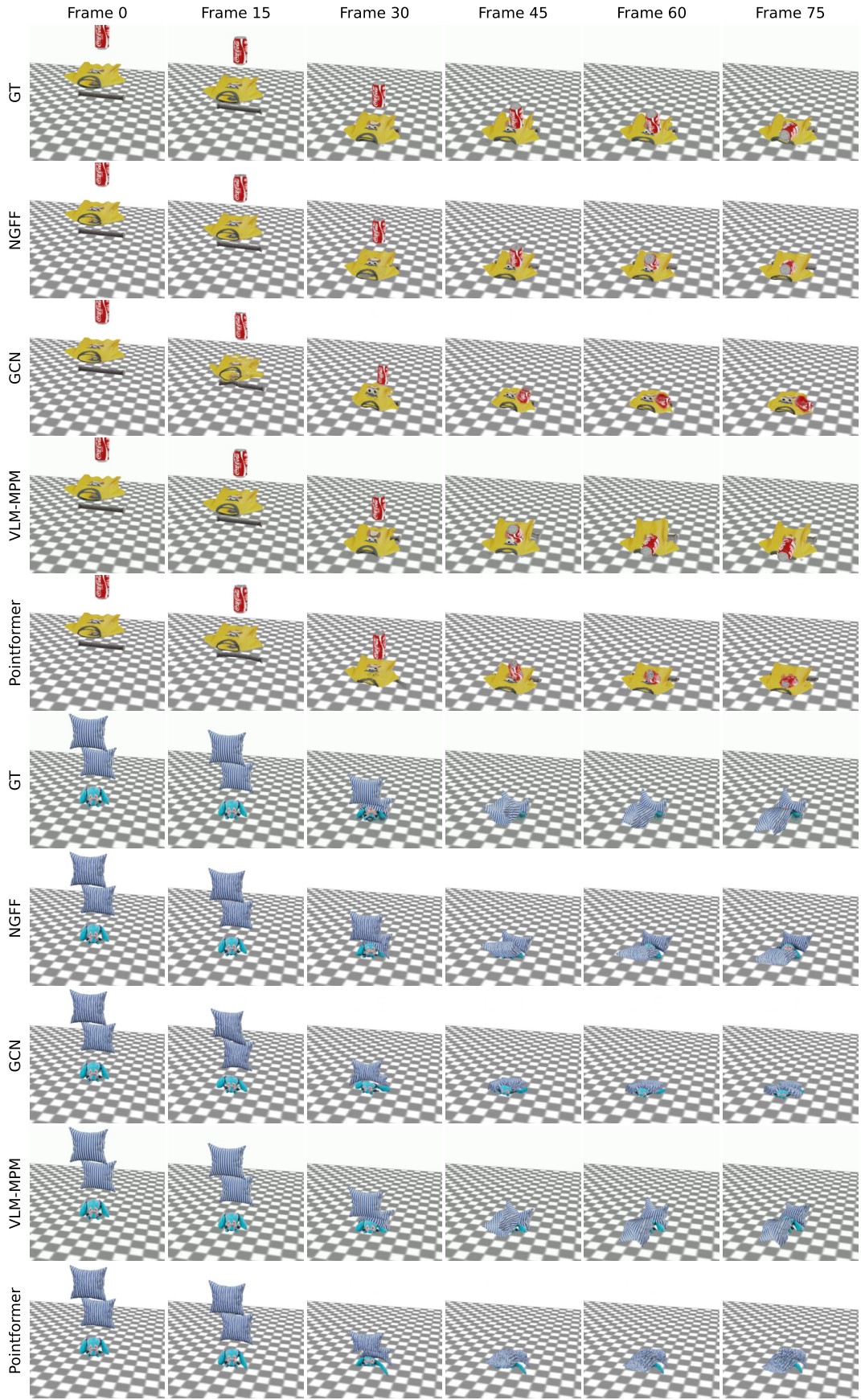

Figure A7: **Dynamic prediction results.**

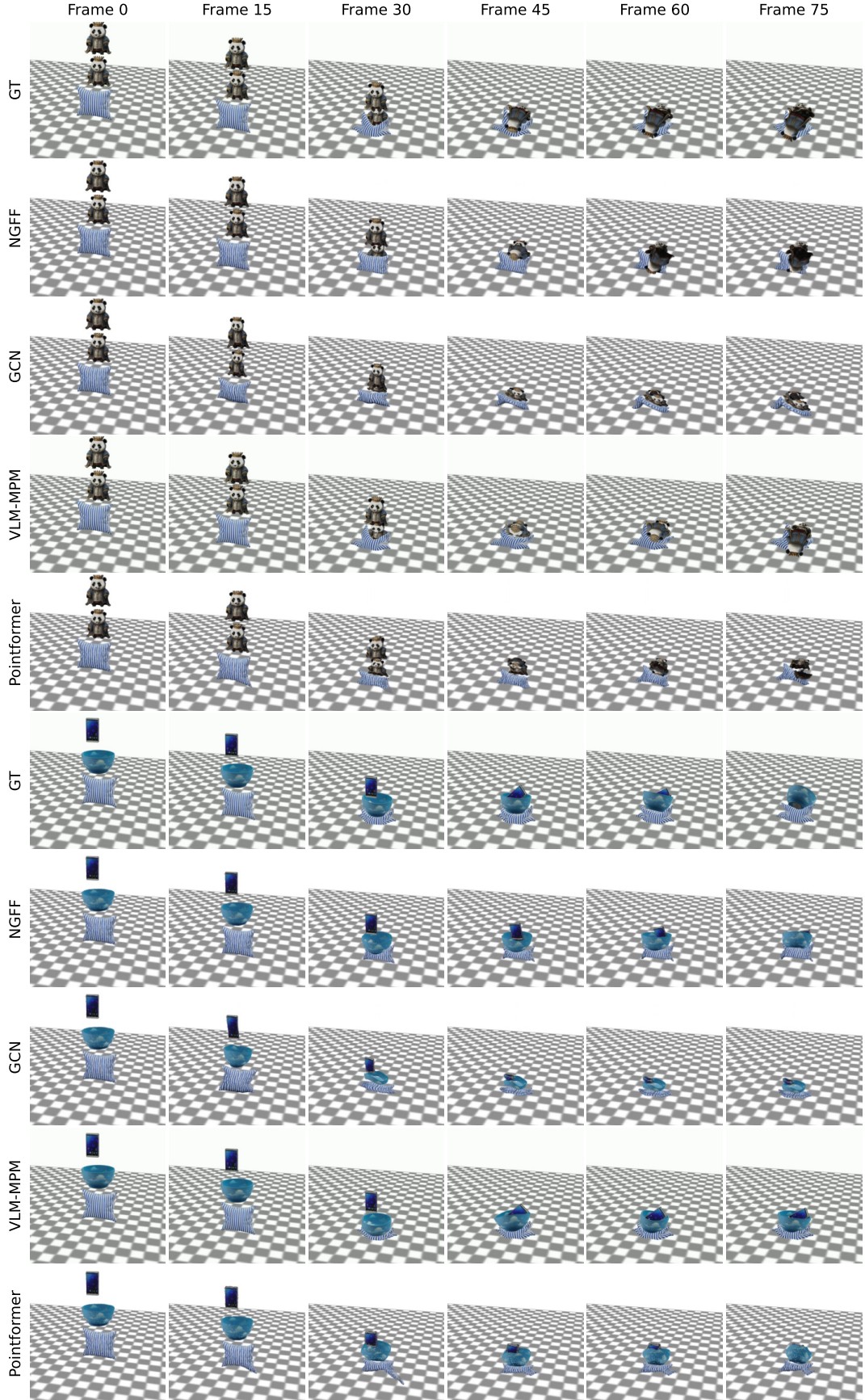

Figure A8: **Dynamic prediction results.**

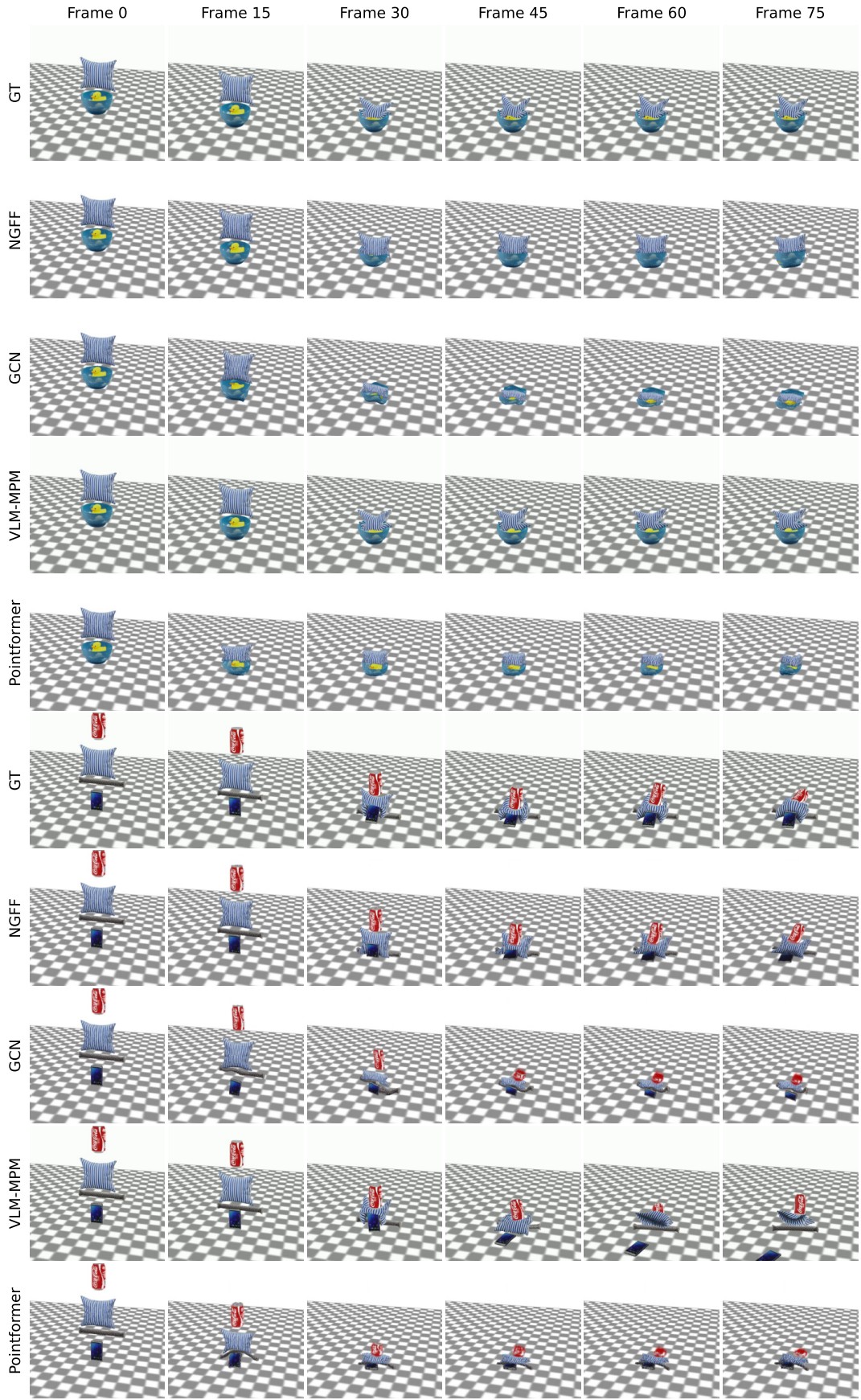

Figure A9: **Dynamic prediction results.**

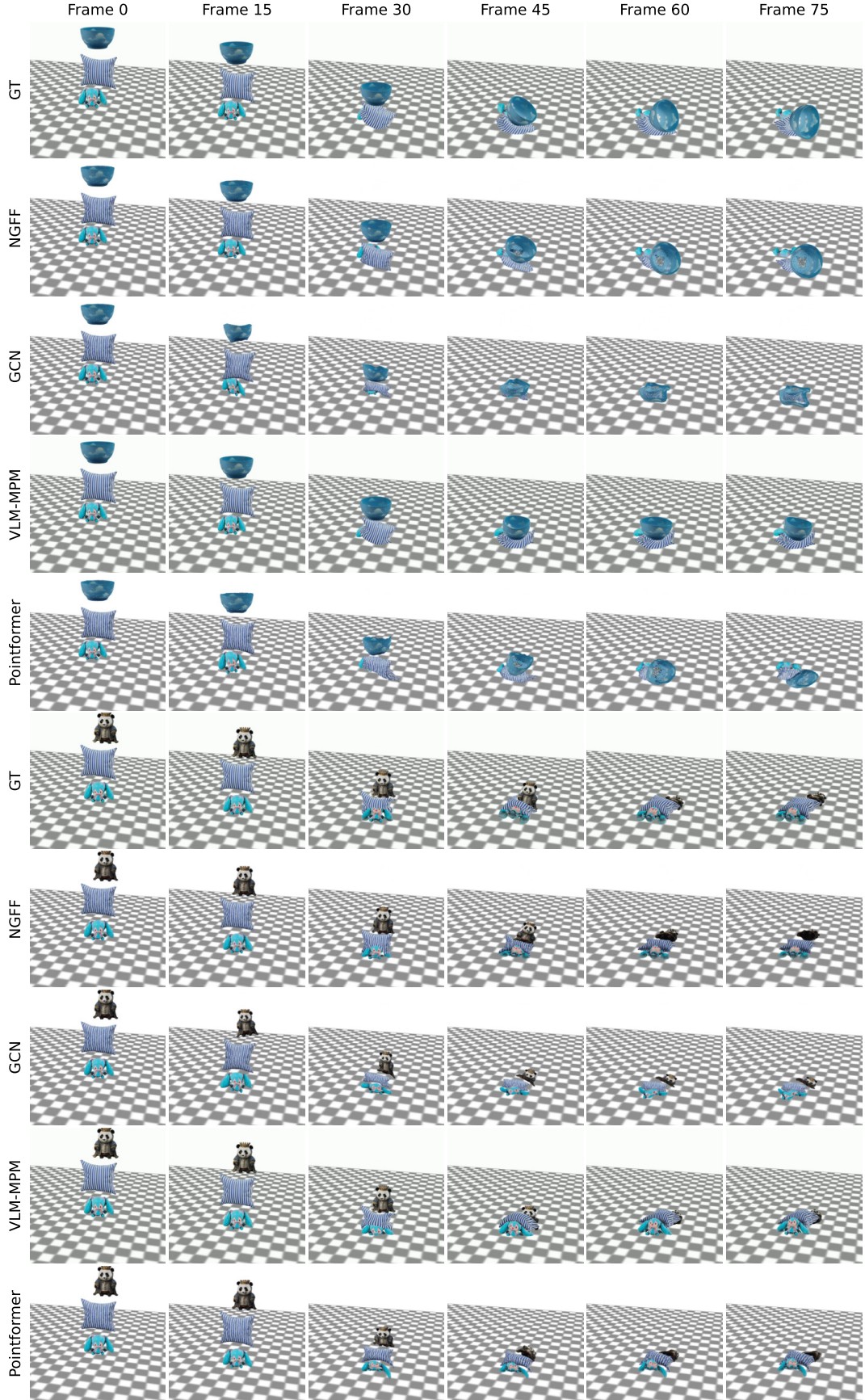

Figure A10: **Dynamic prediction results.**

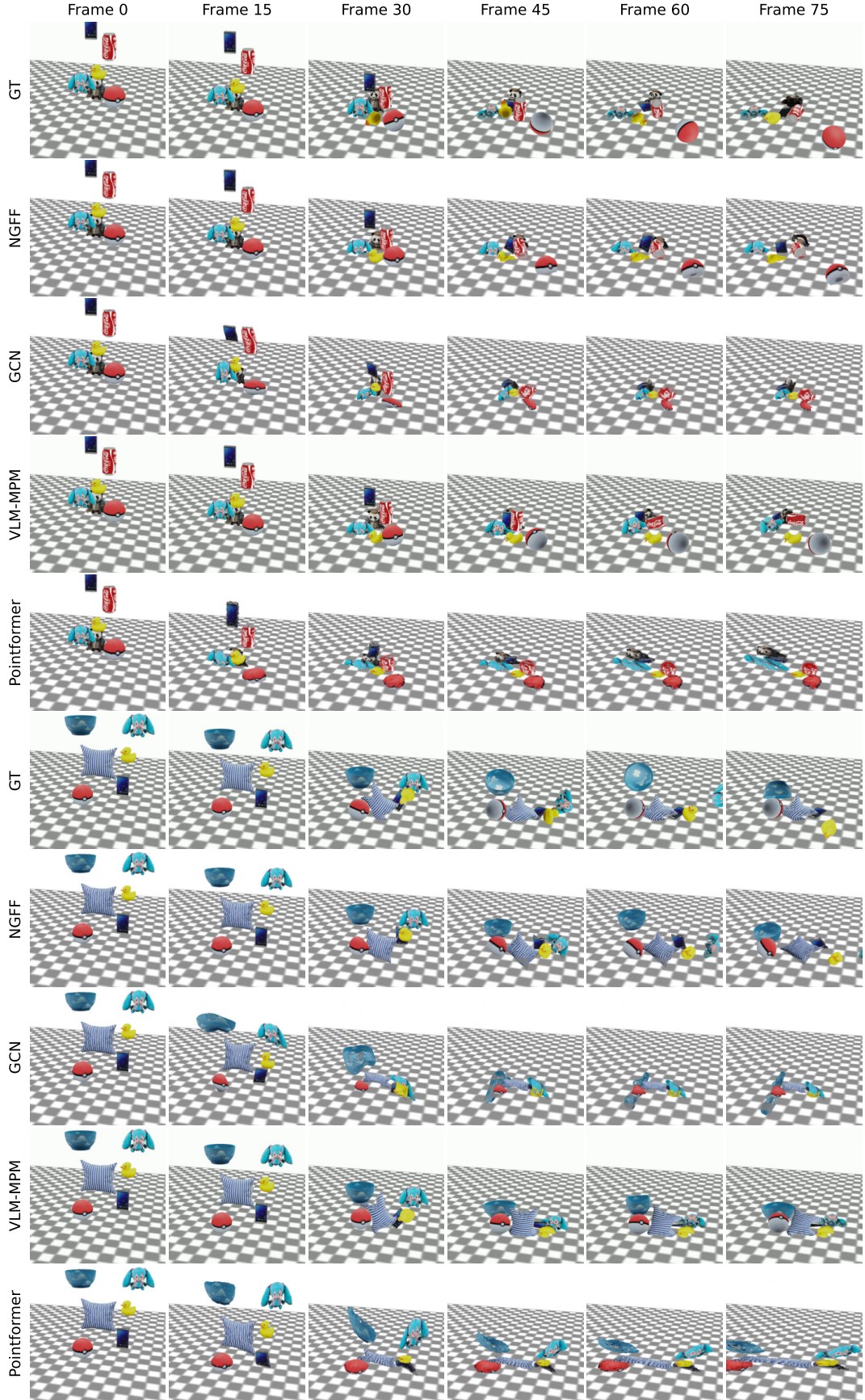

Figure A11: **Dynamic prediction results.**

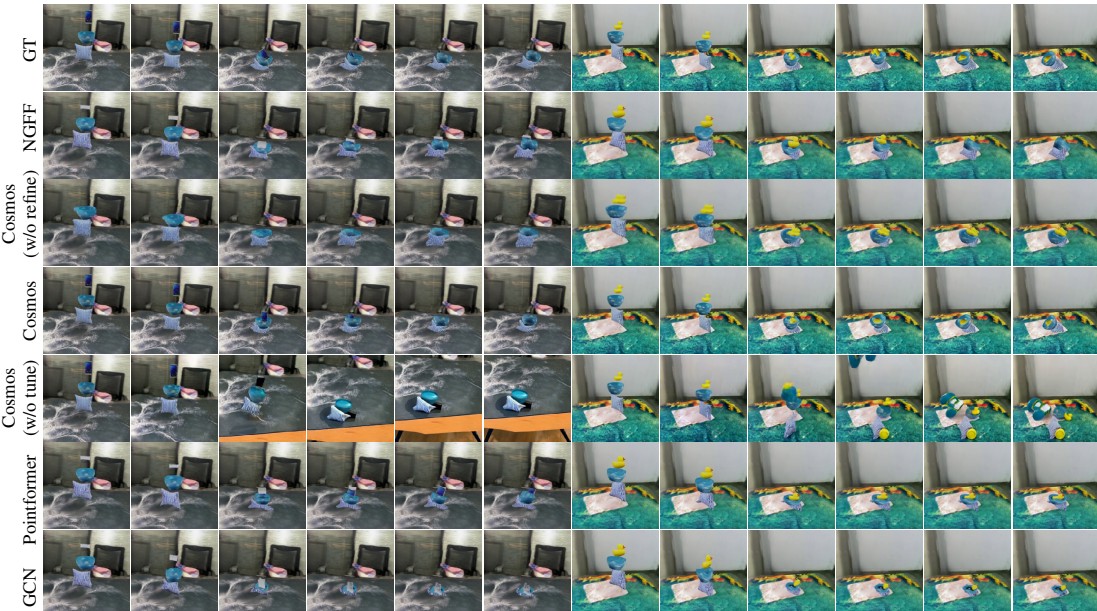

Figure A12: **Video generation results from compositional split.**

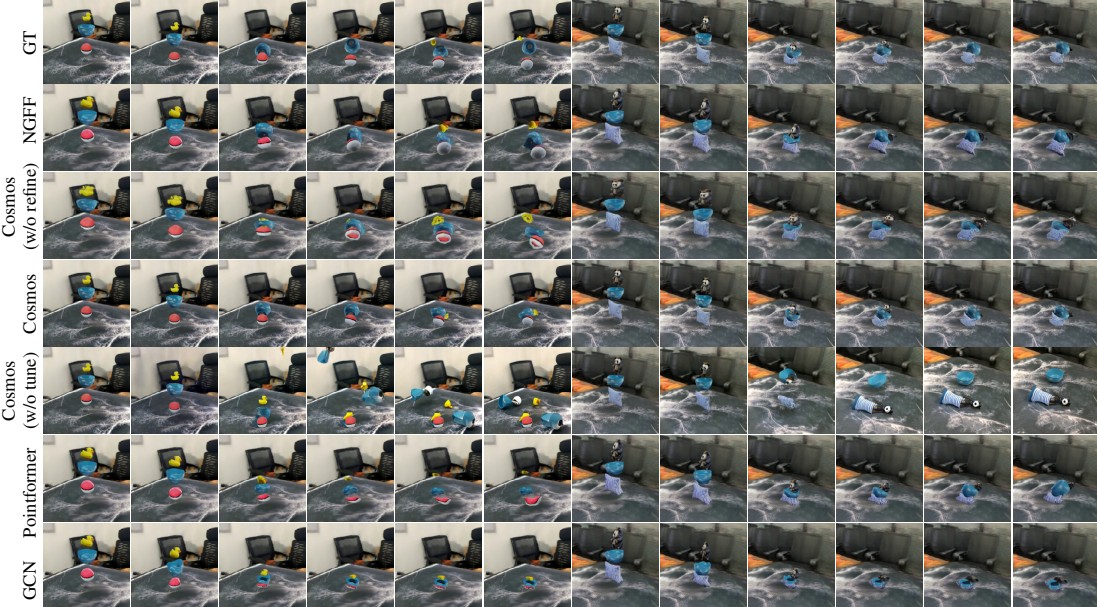

Figure A13: **Video generation results from compositional split.**

## F.2 VIDEO GENERATION

We present additional visualizations of video generation in Figures A12 to A19. We consider monocular reconstruction results in Figure A20 and modeling object of uneven density distribution in Figure A21.

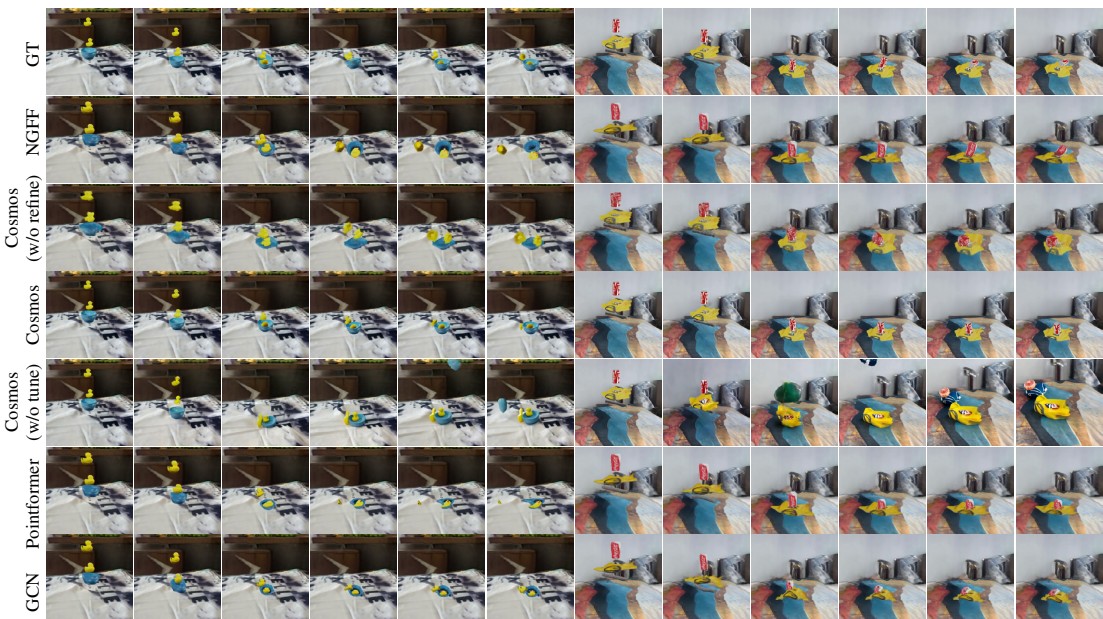

Figure A14: **Video generation results from novel-view split.**

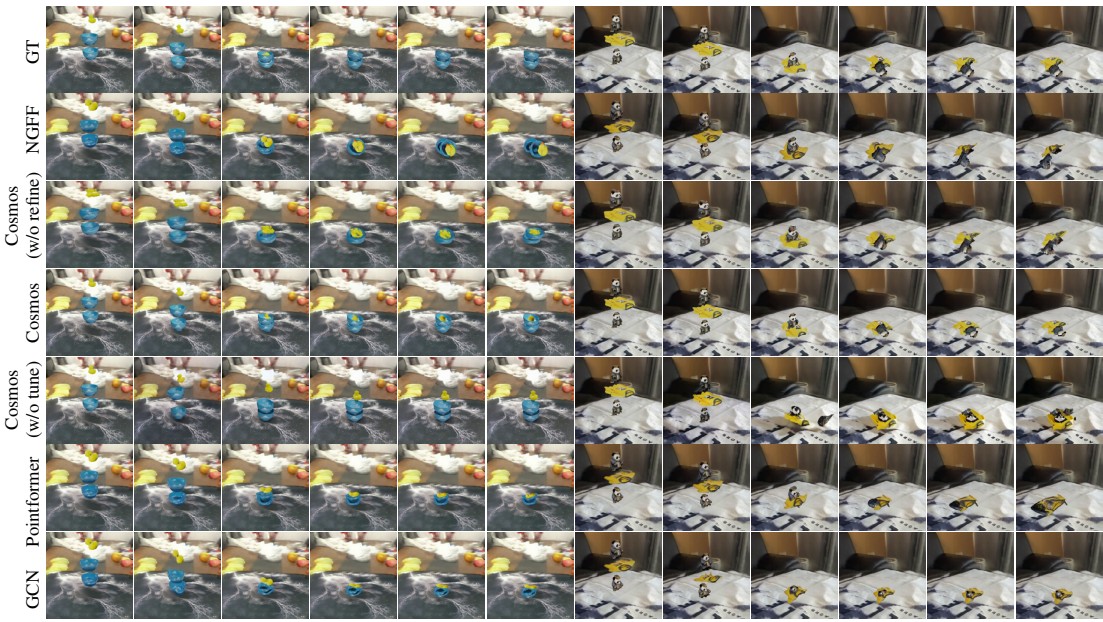

Figure A15: **Video generation results from novel-view split.**

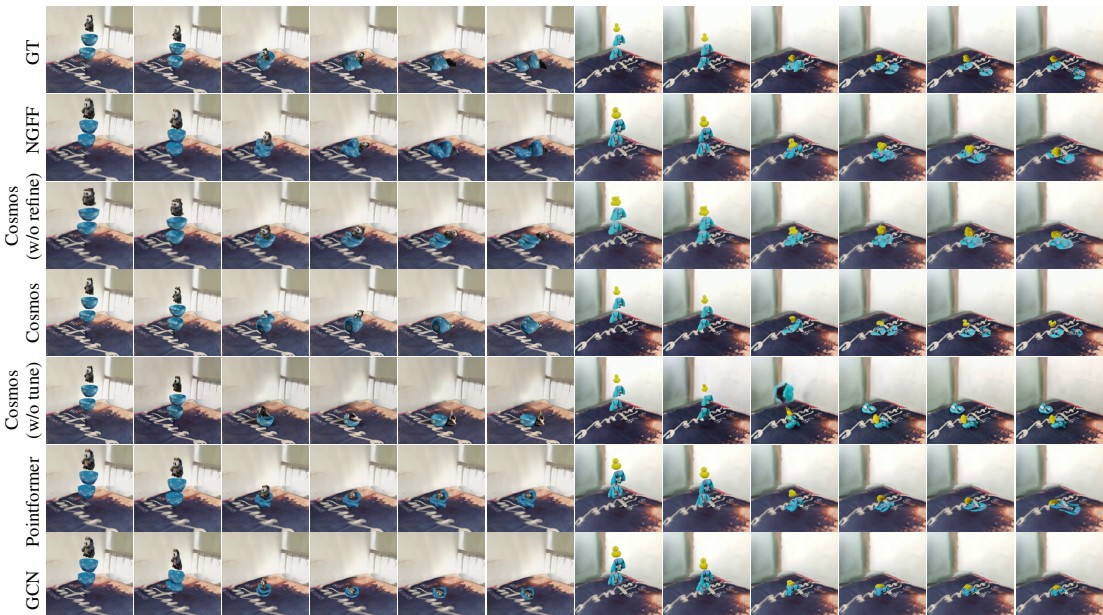

Figure A16: **Video generation results from novel-background split.**

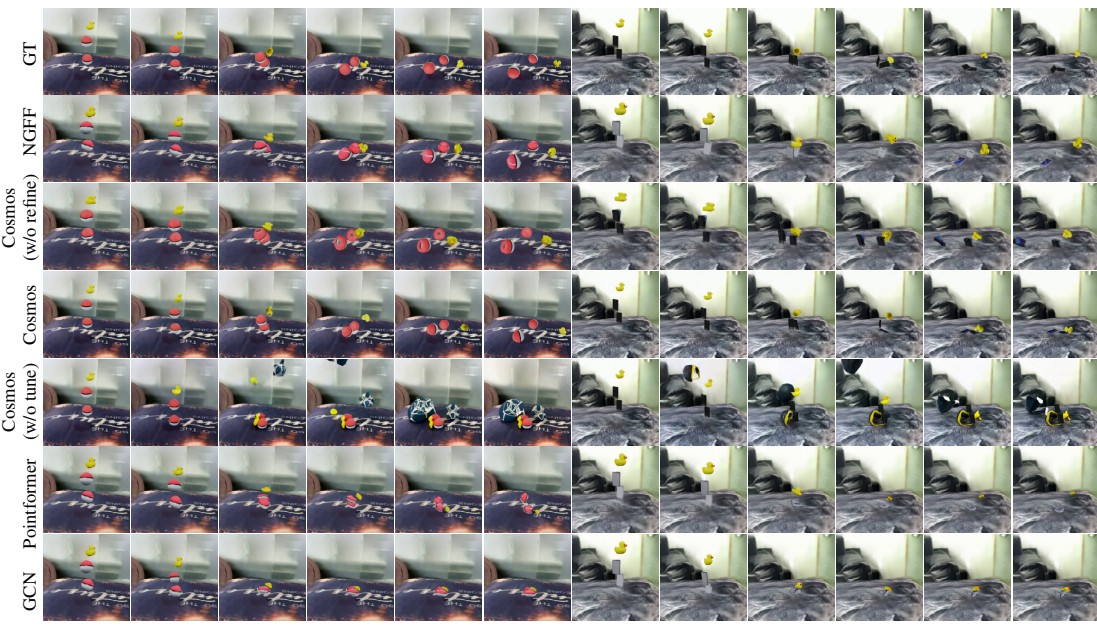

Figure A17: **Video generation results from novel-background split.**

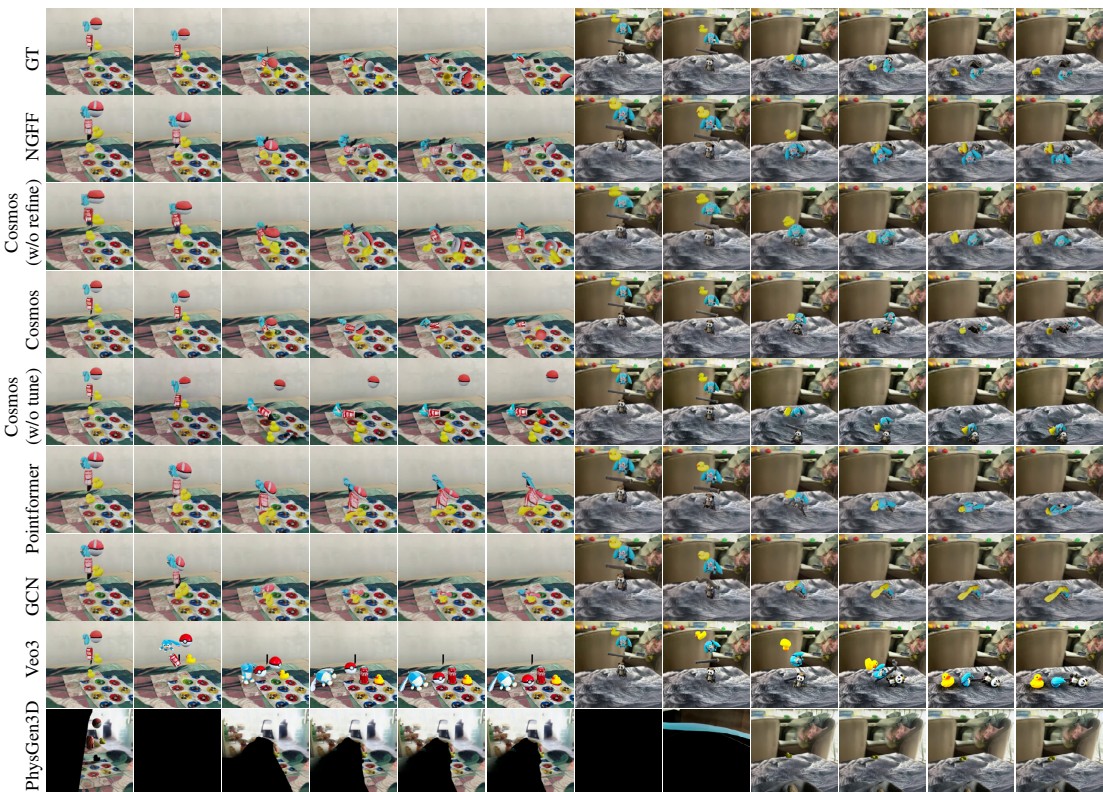

Figure A18: **Video generation results from comprehensive split.**

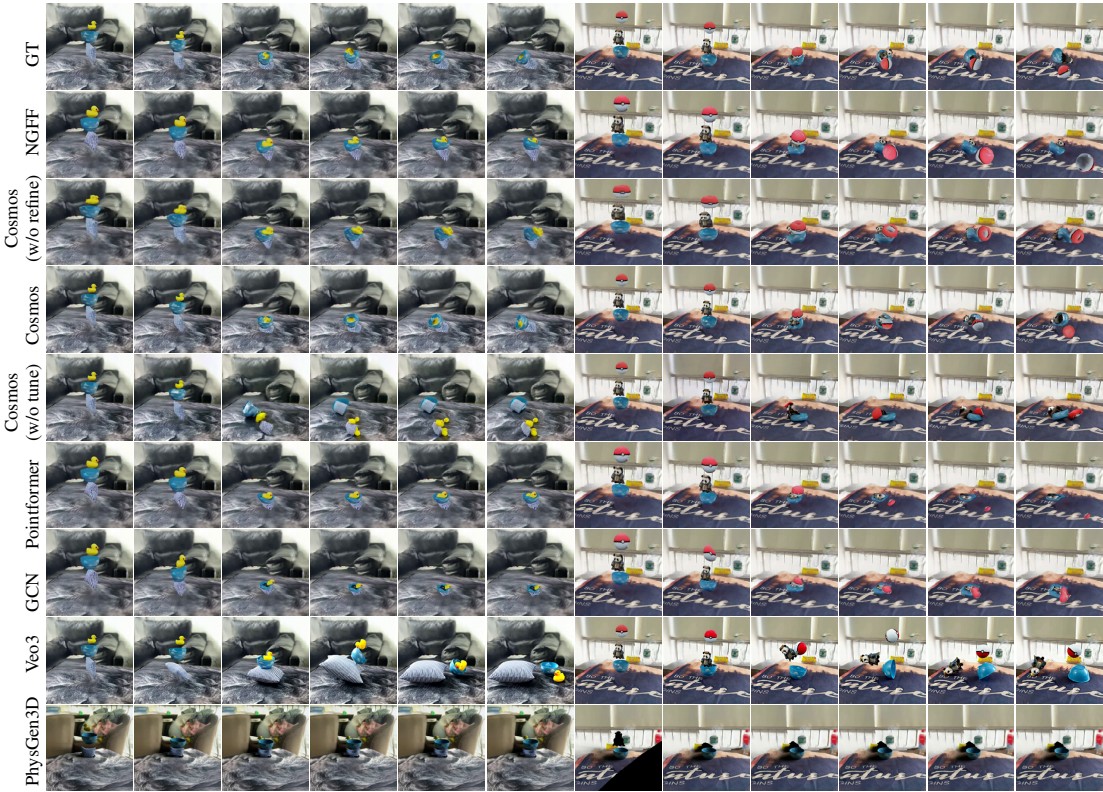

Figure A19: **Video generation results from comprehensive split.**

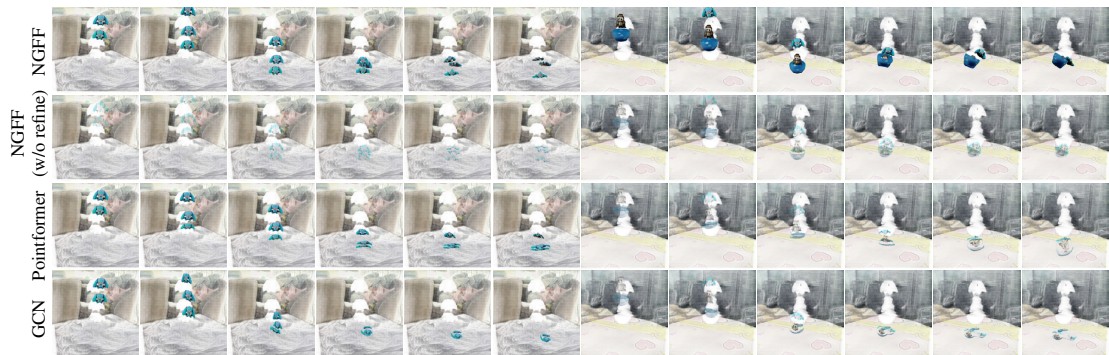

Figure A20: **Video generation results from monocular RGB input.**

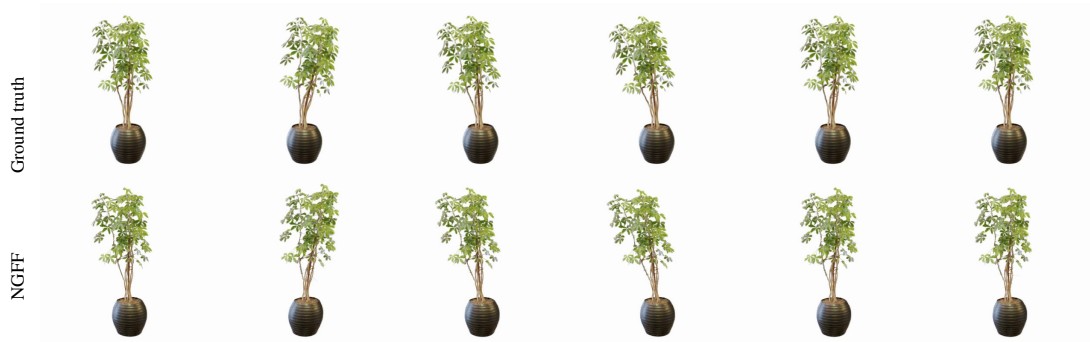

Figure A21: **Dynamic prediction results on non-uniform object modeling.** A shaking ficus with a fixed pot.

