# OpenReview forum: "Learning Physics-Grounded 4D Dynamics with Neural Gaussian Force Fields"
_ICLR.cc/2026/Conference — ICLR 2026 Poster_

### Official Review · Reviewer_QtAo · 2025-10-27

**Soundness:** 3
**Presentation:** 3
**Contribution:** 3
**Rating:** 6
**Confidence:** 3

**Summary:**

This paper introduces NGFF, a framework capable of generating videos with physics-grounded dynamics for scenes containing multi-body interaction. For foreground objects represented by 3D Gaussians, a DeepONet, which consumes the geometric and dynamic attribute of pairwise objects, is employed to predict the received forces and torques in a feed-forward manner. Upon this, local stress fields are predicted to model local deformations. The learned force field is then integrated with an ODE solver to predict object’s motion. In conjunction with the model, they also constructed a dataset containing dynamic Gaussians simulated by MPM for training and benchmark the proposed method. Experimental results show that the proposed feed-forward method can achieve accurate dynamic prediction and demonstrates clear generalization in some aspects.

**Strengths:**

1.	The paper is well-written and easy to follow.
2.	This work shows that force fields in multi-object interactions can be efficiently and accurately predicted by NGFF, with strong positional, temporal and compositional generalization.

**Weaknesses:**

1.	L192 mentions that they leverage 3D generation model to ”address occlusions and invisible parts” from multi-view input, but how to ensure consistency between assets obtained via single (front) view-to-3D generation and the input multi view content in regions invisible in the front view?
2.	In Equation (3), why can the local force field be predicted only from the net force/net torque, contact region and location/velocity of its own each point, which seems under-determined.
3.	The framework does not consider lighting effects induced by motion, which are crucial for realistic video synthesis. This omission weakens its claimed competence as a video generation method.
4.	Fitting the dynamics of only 10 objects already requires a 4T dataset, which appears to suggest a poor data efficiency.

**Questions:**

1.	Does “10 objects” refer to 10 individual instances or 10 categories? If it means categories, how many objects collected for each category, and why not assess generalization to novel instance?
2.	Should generative modeling be considered when scaling to more diverse categories?
3.	L965 says that state descriptors are embedded via an MLP, while L208 mentioned that the state $s(t)$ and $\dot{s}(t)$ containing point cloud and its velocity in $M\times3$. Are these the same “state”? If so, how are the point clouds processed in this MLP?
4.	Some notations in Equation (1) lack clear definition. E.g., $I_{t,k}$ and $\hat{G}$ are not defined in the context. Does this loss aim to enforce consistency of parameters and rendered image between simulated and predicted Gaussian?

---

> ### Author Response · Authors · 2025-11-21
> **Response to reviewer QtAo (Part 1)**
>
> > [W1] L192 mentions that they leverage 3D generation model to "address occlusions and invisible parts" from multi-view input, but how to ensure consistency between assets obtained via single (front) view-to-3D generation and the input multi view content in regions invisible in the front view?
>
> We acknowledge the issue raised by the reviewer: single-view reconstruction does introduce inconsistency in corner cases. Within the 3D community, the generative reconstruction of object Gaussians from multi-view partial observations has been a long-standing challenge. At the time of this paper's submission, we observed that the state-of-the-art generative models available performed even less effectively with multi-view inputs than single-view. So we utilize the single-view DiffSplat model for generative reconstruction in the current manuscript.
>
> However, a recent preprint work, ReconviaGen, which was released after our submission deadline, has addressed this problem well. We have visualized some results at this [anonymous link](https://anonymous.4open.science/r/ICLR26_Rebuttal-C5DF/iclr26_videogen_reconviagen_diffsplat.pdf) using multi-view ReconviaGen instead of DiffSplat as our refinement module backbone. Qualitative comparisons indicate that ReconViaGen helps improve the multi-view consistency of generative reconstruction and makes NGFF's dynamics prediction more accurate.
>
> [1] Chang J, Ye C, Wu Y, et al. ReconViaGen: Towards Accurate Multi-view 3D Object Reconstruction via Generation, arXiv:2510.23306, 2025.
>
> > [W2] In Equation (3), why can the local force field be predicted only from the net force/net torque, contact region and location/velocity of its own each point, which seems under-determined.
>
> We would like to clarify a possible misunderstanding in Equation (3): $x^q(t)$ denotes the point cloud of the entire object, not a single point. Consequently, the local force field predictor takes the full object point cloud and the point cloud velocity together with the contact information as input, which ensures the problem remains well-posed.
>
> > [W3] The framework does not consider lighting effects induced by motion, which are crucial for realistic video synthesis. This omission weakens its claimed competence as a video generation method.
>
> We appreciate your valuable insights. We acknowledge that lightning effects are a crucial factor in video generation quality. Our current model does not take issues such as lighting and shadows into account; rather, it prioritizes the realism of physical interactions, which to some extent compromises the visual realism.
>
> A possible future direction is to combine a generalizable neural simulator with the generative priors introduced by diffusion models [2], or to apply shadow-mapping to the dynamic Gaussian scenes [3]. We have revised the manuscript to include this discussion in section 5.
>
> [2] Alhaija H A, Alvarez J, Bala M, et al. Cosmos-Transfer1: Conditional world generation with adaptive multimodal control, arXiv:2503.14492, 2025.
>
> [3] Chen B, Jiang H, Liu S, et al. PhysGen3D: Crafting a miniature interactive world from a single image, CVPR, 2025.
>
> > [W4] Fitting the dynamics of only 10 objects already requires a 4T dataset, which appears to suggest a poor data efficiency.
>
> Thank you for raising the concern about data efficiency. Actually, our method is data-efficient since we use only 3,200 4D trajectories to train and validate our model. The reported "4T of data" arises from the high temporal resolution and the large number of spatial Gaussian points recorded per trajectory. This is analogous to 3D Gaussian Splatting reconstruction, where a single reconstructed scene can easily reach GB scale.
>
> > [Q1] Does "10 objects" refer to 10 individual instances or 10 categories? If it means categories, how many objects collected for each category, and why not assess generalization to novel instance?
>
> "10 objects" refer to 10 individual instances. In this work, we focus on validating the compositional generalization of the dynamic prediction models in OOD scenes using a predefined list of objects, which follows the convention of previous physical reasoning benchmarks (PHYRE [4], Physion [5], etc.). We will extend the framework to a larger set of object instances and test generalization to novel instances in future work.
>
> [4] Bakhtin A, van der Maaten L, Johnson J, et al. Phyre: A new benchmark for physical reasoning. NeurIPS, 2019.
>
> [5] Bear D M, Wang E, Mrowca D, et al. Physion: Evaluating physical prediction from vision in humans and machines. NeurIPS, 2021.

---

> > ### Author Response · Authors · 2025-11-21
> > **Response to reviewer QtAo (Part 2)**
> >
> > > [Q2] Should generative modeling be considered when scaling to more diverse categories?
> >
> > Thank you for your valuable suggestion. In the dataset synthesis phase, we have already used 3D asset generation models to generate objects, which we believe can help scale to more diverse categories. Besides, for the subsequent video generation phase, we believe that using generative methods will significantly improve the photorealism of the generated videos. We will address this in our future work.
> >
> > > [Q3] L965 says that state descriptors are embedded via an MLP, while L208 mentioned that the state  and  containing point cloud and its velocity in . Are these the same "state"? If so, how are the point clouds processed in this MLP?
> >
> > Point clouds and their local velocities are processed using a PointNet, whereas other object states—such as center of mass, orientation, and object-level velocities—are encoded with MLPs.
> >
> > > [Q4] Some notations in Equation (1) lack clear definition. E.g.,  and  are not defined in the context. Does this loss aim to enforce consistency of parameters and rendered image between simulated and predicted Gaussian?
> >
> > Sorry for the confusion. You are correct in your assumption about their purpose; they are designed to enforce both image and parameter consistency between the ground truth and our predictions.
> > We will clarify these terms in our revision:
> >
> > $\hat{I}_{t,k}$ : The predicted and rendered image from viewpoint $k$ at time $t$ .
> >
> > $I_{t,k}$ : The ground-truth image from viewpoint $k$ at time $t$ .
> >
> > $\hat{\mathcal{G}}_t$ : The predicted set of Gaussians at time $t$ .
> >
> > $\mathcal{G}_t$ : The ground-truth set of Gaussians (from the MPM simulation) at time $t$ .

---

> ### Comment · Reviewer_QtAo · 2025-11-24
>
> I have read the author’s response and appreciate their efforts. However, despite the clarification of specific details (Q1), most of my concerns get nothing adequately addressed other than semantic distractions. Moreover, the clarification also deepens my concern about the method’s generalization to novel instance and diminishes the value of the claimed faster simulation.
>
> **[W1]** I don’t want to be distracted and belabor whether author’s discussion around 3D generation/reconstruction in rebuttal is correct and thorough or why inconsistency in half of object (i.e. the whole back side) could be dismissed as “corner case”. The issue I raised in my original comment is that, since the proposed pipeline only uses single view-to-3D generation and cannot really guarantee the consistency with other views, the stated reliance on multi-view input and motivation to “address occlusions and invisible parts from inputs” are inaccurate and misleading. But in fact, despite a large part of text and half of Figure 2 being used to present it, the object-centric reconstruction appears to be disconnected with the whole framework and not really involved in most of experiments where oracle 3D Gaussians is available, nor does it provide any technical novelty. This component seems to apply only for constructing training data, rather than being directly applicable to unseen novel objects on-the-fly as implied because the subsequent NGFF and thus the entire framework is restricted to a predefined fixed set of 10 objects with no demonstrated ability to generalize beyond them. Moreover, the lack of ability to generalize to novel objects also undermines the claimed advantages of faster simulation--if additional simulation and training are required to handle a new given object, it is somewhat unfair to compare the efficiency with simulation-based counterparts without accounting for that time.
>
> **[W2]** Sorry for the confusion if it is not a deliberate distraction to sidestep the original issue. In my previous comment, “location/velocity of its own each point” does refer to the location and velocity of each point in its whole point cloud. I am curious how this could be interpreted as a single point, considering that the “single point of an object” is not defined in the current context. My concern is that, even given the full point cloud of an object, mask of contact region, and its net force/torque, the local force field remains under-determined. For example, consider a square panel with pressure applied at its center and each corner. It is easy to construct different force allocations among these 5 points that yield the same net force and torque while producing different local force fields. In all these cases, the contact-area mask and the location and velocity of each point remain identical.
>
> **[W3]** While I appreciate the author’s discussion of possible future directions, it does not mitigate my concerns about its current incompetence in video generation as well as its overclaim about photorealism (L257).
>
> **[W4&Q1]** Obscuring the definition of data efficiency does not helps resolve the existing issues. If fitting 10 predefined objects already requires a 4T dataset, the proposed approach would be practically infeasible to scale and generalize beyond the current toy setting. The current framework seems to only memorizes the physical properties of a fixed set of objects and interaction dynamics among them. Now the experiments of compositional generalization only vary the number of interacting objects, there is even no evidence showing that the framework can generalize to combinations of objects unseen together during training, e.g., if pillow and ball never appear together in any training scene, can the model still predict their interaction in test time? If not, the required dataset size would grow quadratically with the number of objects.
>
> **[Q3]** I acknowledge that the description in supplementary material is clear enough. Actually, my comment is intended to remind the inconsistent use of specific terminology: lines 194–200 introduces three “states,” while none of them matches the “state descriptor” defined in supplementary material, which somewhat compromises the clarity of the manuscript.
>
> **[Q4]** Since the ground-truth Gaussian states are already available as supervision, is an additional photometric loss really necessary? The photometric loss may exhibit undesirable landscape wrt. some attributes like position under large motion.

---

> > ### Author Response · Authors · 2025-11-25
> > **Response to reviewer QtAo (Part 1)**
> >
> > We sincerely thank the reviewer for the continued engagement and the insightful feedback. We apologize if we misunderstood your points and appreciate the opportunity to further clarify the details.
> >
> > > since the proposed pipeline only uses single view-to-3D generation and cannot really guarantee the consistency with other views, the stated reliance on multi-view input and motivation to "address occlusions and invisible parts from inputs" are inaccurate and misleading.
> >
> > 1. **Reliance on multi-view input**: In our initial submission, the input used for feed-forward scene Gaussian reconstruction was multi-view, while the input for subsequent 3D object refinement module was single-view. We will further clarify their difference in our revision.
> >
> > 2. **Address occlusions and invisible parts from inputs**: Our motivation for introducing the 3D object refinement module was aimed at inpainting areas such as the unseen underside of a ball or garments partially occluded by overlying objects to perform more accurate physics predictions (e.g., a ball missing its lower half would result in object interpenetration). For these scenarios, standalone multi-view reconstruction is insufficient; even with a surround-view setup, objects in the scene may still lack complete geometry due to factors such as camera elevation and occlusions. Under these conditions, we think it is necessary to incorporate generative object priors.
> >
> > 3. **Why use single-view-to-3D?** In our initial submission, we generated complete 3D assets from single-view images because current foundation models generally perform better on single-view–to–3D than on multi-view–to–3D generation, despite the multi-view inconsistencies that single-view methods can introduce. Fortunately, NGFF framework is designed to be compatible with various 3D refinement backbones, and we obeserve that leveraging more advanced foundation models, such as multi-view ReconViaGen, can mitigate these issues.
> >
> > We will update our documentation to make these statements clearer.
> >
> > > the object-centric reconstruction appears to be disconnected with the whole framework and not really involved in most of experiments where oracle 3D Gaussians is available. This component seems to apply only for constructing training data, rather than being directly applicable to unseen novel objects on-the-fly as implied.
> >
> > We wish to clarify that the object-centric reconstruction module was not used to "construct training data". Instead, it is employed within the video generation setting to recover object Gaussians from RGB inputs and predict their dynamics. In our video generation and real-world experiments (section 4.3 and 4.4)—where object configurations and positions differ from those in the dynamics training—we do not have access to oracle 3D Gaussians. In this setting, we rely solely on pure multi-view RGB inputs. The model itself needs to handle noisy reconstruction results and infer the subsequent Gaussian dynamics. Furthermore, for the object-centric reconstruction module only, since we utilized frozen pretrained foundation models, it can naturally generalize to unseen objects on-the-fly.
> >
> > > the entire framework is restricted to a predefined fixed set of 10 objects with no demonstrated ability to generalize beyond them. Moreover, the lack of ability to generalize to novel objects also undermines the claimed advantages of faster simulation
> >
> > We recognize that additional efforts are needed to test the generalization to new objects. Actually, our results from real-world experiments (section 4.4) provide some evidence of the model's ability to generalize to unseen objects. For instance, it successfully reconstruct a teddy bear and predict its dynamics that was not part of our training data, as well as a cola can and a duck that differ in shape from the examples used during training. With those unseen objects, our NGFF outperforms other baseline neural models in dynamic predictions. We'd like to perform a more thorough test of this generalization in the future.

---

> > > ### Author Response · Authors · 2025-11-25
> > > **Response to reviewer QtAo (Part 2)**
> > >
> > > > it is somewhat unfair to compare the efficiency with simulation-based counterparts without accounting for that time.
> > >
> > > We agree that simulation-based methods offer greater generality and flexibility compared to neural dynamics methods. The intention of NGFF is not to replace physics simulation methods, but to provide a faster "neural proxy" for tasks requiring high real-time performance, such as deploying on robots. Furthermore, when inferring object dynamics purely from RGB inputs, neural dynamics tend to be more robust to noisy reconstruction results than traditional simulators. Besides, simulation-based methods typically rely on Vision Language Models for parameter estimation [1] or perform differentiable simulation leveraging video diffusion priors [2] —steps that can be inaccurate or time-consuming.
> > >
> > > [1] Chen B., Jiang H., Liu S., Gupta S., Li Y., Zhao H., Wang S. PhysGen3D: Crafting a Miniature Interactive World from a Single Image. CVPR 2025.
> > >
> > > [2] Lin Y., Lin C., Xu J., Mu Y. OmniPhysGS: 3D Constitutive Gaussians for General Physics‑Based Dynamics Generation. ICLR 2025.
> > >
> > > > [W2] Sorry for the confusion if it is not a deliberate distraction to sidestep the original issue. In my previous comment, "location/velocity of its own each point" does refer to the location and velocity of each point in its whole point cloud. I am curious how this could be interpreted as a single point, considering that the "single point of an object" is not defined in the current context. My concern is that, even given the full point cloud of an object, mask of contact region, and its net force/torque, the local force field remains under-determined. For example, consider a square panel with pressure applied at its center and each corner. It is easy to construct different force allocations among these 5 points that yield the same net force and torque while producing different local force fields. In all these cases, the contact-area mask and the location and velocity of each point remain identical.
> > >
> > > We sincerely apologize for the misunderstanding in our previous response, where we misinterpreted the concern regarding the uniqueness of the inverse problem as a question about the definition of the input features.
> > >
> > > We appreciate the opportunity to clarify a potential notational ambiguity in $\mathbf{F}^{\text{global}(\mathbf{z}^q(t))}$ between Eq. (2) and Eq. (3). We acknowledge the validity of the example you constructed; distinguishing the force distribution across five contact points is indeed impossible if one relies solely on identical net force and net torque. However, in our actual code implementation, the input to the Local Stress Net in Eq. (3) is actually a latent vector derived from the DeepONet, which is different from the net force and net torque used for ODE integration in Eq.(2). This latent vector is not a simple summation in Newtonian space; rather, it encodes geometric information from the acting object, enabling the StressNet to decode it into distinct force distributions across different contact points. Given that PointNet serves as a universal approximator for object point clouds, the NGFF framework theoretically shares the same upper bound of expressiveness as particle-based dynamics.
> > >
> > > To avoid misunderstanding caused by the reuse of notations, we have revised the manuscript to explicitly indicate $\mathbf{F}^{\text{latent}}(\mathbf{z}^q(t))$ in Eq. (3) as a latent vector and split the output of the DeepONet into two parts: $\mathbf{F}^{\text{global}}(\mathbf{z}^q(t))$ and $\mathbf{F}^{\text{latent}}(\mathbf{z}^q(t))$ in Eq.(1).
> > >
> > > > [W3] While I appreciate the author’s discussion of possible future directions, it does not mitigate my concerns about its current incompetence in video generation as well as its overclaim about photorealism (L257).
> > >
> > > We acknowledge the degradation in photorealism of our method compared to SOTA video generation models, especially in lighting effect. However, we think the evaluations from VLMs and 60 human participants in Table 2 indicates that the overall photorealism is accepteble and comparable with SOTA video generation models in certain splits. We will revise the manuscript to remove potential overclaim.

---

> > > > ### Author Response · Authors · 2025-11-25
> > > > **Response to reviewer QtAo (Part 3)**
> > > >
> > > > > [W4&Q1] Obscuring the definition of data efficiency does not helps resolve the existing issues. If fitting 10 predefined objects already requires a 4T dataset, the proposed approach would be practically infeasible to scale and generalize beyond the current toy setting. The current framework seems to only memorizes the physical properties of a fixed set of objects and interaction dynamics among them. Now the experiments of compositional generalization only vary the number of interacting objects, there is even no evidence showing that the framework can generalize to combinations of objects unseen together during training, e.g., if pillow and ball never appear together in any training scene, can the model still predict their interaction in test time? If not, the required dataset size would grow quadratically with the number of objects.
> > > >
> > > > We would like to clarify that the original 4T dataset contains high-resolution spatial-temporal Gaussian data for all scenes. We reserved them for potential downstream tasks (e.g., to render the 216K high-quality videos required for fine-tuning the video generation model). However, we only used a small subset of this data during training. Specifically, to ensure a manageable training data size, we employed downsampling techniques. We used **Farthest Point Sampling (FPS)** during preprocessing. While the raw Gaussian point counts for each object range from 8,000 to 200,000, we reduced each object to just 2,048 points. As a result, the effective training data dimensions are [batch_size=2700, time=80, num_objects=3, num_points=2048, xyz_dimensions=3], totaling approximately **14.8 GB**. In the rendering process, we utilize the complete Gaussians with their attributes only in the initial frame. We then perform upsampling based on the predicted motion of these 2,048 keypoints while interpolating other Gaussian parameters, such as scale and rotation, to reconstruct the full video sequence. We will provide a reduced dataset that retains only the essential information for training in order to make it more applicable and practical for scaling up.
> > > >
> > > > Regarding the generalization to unseen objects and their combinations, we kindly refer you to the response to [W1].
> > > >
> > > > > [Q3] I acknowledge that the description in supplementary material is clear enough. Actually, my comment is intended to remind the inconsistent use of specific terminology: lines 194–200 introduces three "states," while none of them matches the "state descriptor" defined in supplementary material, which somewhat compromises the clarity of the manuscript.
> > > >
> > > > We will remove "state descriptor" and directly use "center of mass, orientation angles, linear and angular velocities".
> > > >
> > > > > [Q4] Since the ground-truth Gaussian states are already available as supervision, is an additional photometric loss really necessary? The photometric loss may exhibit undesirable landscape wrt. some attributes like position under large motion.
> > > >
> > > > We agree with your point. We observe that in the presence of ground truth, photometric loss is less critical for the dynamics model. While our approach theoretically allows for fine-tuning based on image space loss when there is no groundtruth gaussian states available, we recognize this term may be misleading and have therefore removed it from the manuscript.

---

### Official Review · Reviewer_sf96 · 2025-10-29

**Soundness:** 3
**Presentation:** 3
**Contribution:** 3
**Rating:** 6
**Confidence:** 3

**Summary:**

This paper introduces Neural Gaussian Force Field (NGFF) to address the challenge of predicting physical dynamics from raw visual data. It learns an implicit force field to drive the temporal evolution of 3D Gaussian Splatting, thereby modeling 4D dynamics. Experimental results demonstrate that NGFF achieves state-of-the-art performance in the 4D generation of collision motion in both simulation and real-world cases.

**Strengths:**

1. The idea of using an implicit force field, rather than an explicit physics engine, to govern 3D GS evolution is highly innovative. It effectively tackles the critical bottlenecks of large computational overhead and insufficient robustness inherent in tightly coupling 3D GS with physical simulation.
2. The paper provides extensive quantitative and qualitative results across diverse and challenging evaluations, which demonstrate NGFF's effectiveness in both 4D dynamic prediction and the more challenging real-world simulation.

**Weaknesses:**

1. Limited Performance: Since the model is primarily trained on data generated by the Material Point Method (MPM) simulation, its learned dynamics are inherently limited by the accuracy, fidelity, and approximations of the underlying MPM solver. This raises a concern that the performance of the trained NGFF model is limited to the quality of the synthetic ground truth.
2. Generalization to Unseen Physics: While NGFF performs excellently on the provided datasets, its ability to generalize to unseen material properties (e.g., how a model trained only on rigid and soft bodies handles sand or fluids) or unseen complex constraints (e.g., complex joints or hinges) remains unclear. This is a crucial factor for real-world applicability.
3. Missing Implementation Details: Modeling and training details are not clear enough, making this paper hard to follow. For example, what attributes of Gaussian kernels need to be supervised, and how is L' calculated in Eq.1? What is $s$ in Eq.6, and how to convert it to the state of Gaussian kernels?

**Questions:**

My questions are mainly based on the above weaknesses:
1. Do the predictions from NGFF perform better compared to the MPM simulation results? Is it possible to train the model with real captured data?
2. The predicted simulation in the paper is limited to the collision of rigid or soft body objects. Can this method be extended to more complex physical motions?

---

> ### Author Response · Authors · 2025-11-21
> **Response to reviewer sf96**
>
> > [W1] its learned dynamics are inherently limited by the accuracy, fidelity, and approximations of the underlying MPM solver
>
> We appreciate the concerns regarding the fidelity gap between simulated and real-world data.
>
> Collecting and processing real-world videos into trainable components, such as 3D representations with point correspondence, presents a significant, **costly scalability barrier**. Therefore, the primary motivation for employing MPM simulation was to enable **easy, fast, and cost-effective** generation of physics-grounded 4D data at scale. This paradigm is crucial for research in dynamic prediction and robotics, where models are often trained in simulation before transfer to the real world.
>
> To ensure the highest possible fidelity, we utilized an advanced GPU-accelerated MPM engine, featuring **high-resolution** grids and simulation steps, along with objects and backgrounds **reconstructed from real-world scenes** (e.g., WildRGBD). For the first step, we focus on complex object interactions on a table, so we choose a box area for simulation. While this pipeline may not eliminate the sim-to-real gap entirely, it provides a highly scalable and flexible platform for rapidly validating the effectiveness of our method. We believe further improvement can be achieved with the progress in both physics simulation (MPM) and 3D reconstruction techniques (3DGS), and future work will focus on further enhancing the quality of the simulated data.
>
> Besides, another reasonable next step to alleviate sim-to-real gap is to study how to transfer the simulation data into the real-world by fine-tuning the pretrained model on a small set of real data.
>
> > [W2] While NGFF performs excellently on the provided datasets, its ability to generalize to unseen material properties
>
> We acknowledge that NGFF does not generalize well to extremely out-of-distribution object interactions, which remains a fundamental challenge for all neural dynamics models. Nevertheless, NGFF’s explicit force modeling provides strong data efficiency, allowing it to adapt to new scenarios from only a limited number of examples—without requiring internet-scale datasets, in contrast to typical video generation models.
>
> > [W3] Missing Implementation Details
>
> We will clarify these details in our revision. For your questions: Only the Gaussian kernel positions are supervised. The covariances at future time steps are computed using a rotation matrix derived from a local rigidity constraint applied at the initial step. All other attributes (i.e. opacity, color, and spherical harmonics) are kept unchanged from the initial configuration, without noticeably affecting rendering quality. L' in Eq.1 is calculated as the MSE loss of predicted and ground-truth Gaussian kernel positions. The $\mathbf{s}$ and $\dot{\mathbf{s}}$ . In Eq. 6 refers to the zeroth-order states (e.g. point cloud) and first-order states (e.g. point cloud velocity), respectively.
>
> > [Q1] Do the predictions from NGFF perform better compared to the MPM simulation results? Is it possible to train the model with real captured data?
>
> Since our dataset is generated by MPM simulation, we compare the model's prediction results against the MPM ground truth in the experiments. However, we would like to clarify that MPM simulation requires accurate point cloud modeling and parameter estimation of the object's material, which are difficult to obtain with only RGB observations. NGFF is more robust to noisy point clouds and offers significant acceleration compared to MPM simulation. Therefore, in typical scenarios where **only RGB input is available**, NGFF can perform better compared to MPM.
>
> In the Dynamics prediction section, our paper includes experiments on using Vision Language Models to predict object materials for simulation, demonstrating that inaccurate parameter estimation significantly affects the accuracy of dynamics prediction. For your convenience, we include the results below.
>
> | Model | Spatial RMSE ↓ | Temporal RMSE ↓ | Compositional RMSE ↓ |
> | :--- | :--- | :--- | :--- |
> | VLM-MPM | 0.306 | 0.328 | 0.358 |
> | **Our NGFF** | **0.082** | **0.107** | **0.104** |
>
> > [Q2] The predicted simulation in the paper is limited to the collision of rigid or soft body objects. Can this method be extended to more complex physical motions?
>
> In the supplementary experiments, we added an example using an object with a non-uniform mass distribution (a shaking ficus with soft upper body but rigid lower body). Please see the results in our revised pdf (Appendix Figure A21). It shows that our model can learn its dynamic pattern with a small MSE loss of 2.54e-4. Our modeling is theoretically applicable to various materials, including fluid models and granular models, which we will explore in future work.

---

### Official Review · Reviewer_jh1Y · 2025-11-02

**Soundness:** 3
**Presentation:** 3
**Contribution:** 3
**Rating:** 8
**Confidence:** 3

**Summary:**

The NGFF framework integrates 3D Gaussian perception with Neural Gaussian Force Fields to generate interactive, physically realistic 4D videos from multi-view inputs. It models dynamics by learning an explicit force field integrated via an ODE solver, achieving two orders of magnitude faster simulation than previous Gaussian methods.

**Strengths:**

The NGFF framework integrates 3D Gaussian perception and Neural Gaussian Force Fields to generate interactive, physically realistic 4D videos. It models dynamics by learning an explicit force field, achieving two orders of magnitude faster simulation. It also introduces the GSCollision dataset.

**Weaknesses:**

The NGFF framework currently relies on multi-view inputs for reliable 3D Gaussian reconstruction, needing extension to monocular or partial observations to match human ability. The benchmark covers only 10 representative objects, necessitating scaling to far more diverse materials and articulated structures. Additionally, visual quality is sometimes slightly lower than SOTA video generation models due to 3D reconstruction error.

**Questions:**

How will you extend NGFF to robustly predict 4D dynamics from monocular or partial RGB inputs?

---

> ### Author Response · Authors · 2025-11-21
> **Response to reviewer jh1Y**
>
> > [W1] relies on multi-view inputs for reliable 3D Gaussian reconstruction
>
> Thank you for the insightful suggestion. We have added extra results using monocular RGB as input. Please see the detailed reply to Q1.
>
> > [W2] The benchmark covers only 10 representative objects
>
> Since our work focuses on compositional generalization across different scenes, we restrict our study to a small set of representative objects. Scaling the number of objects to the order of thousands or more would incur substantially higher computational costs. We plan to expand the dataset to include richer and more diverse physical scenes in future work. In addition, we have included a set of experiments involving objects with articulated structures (e.g., a shaking ficus consisting of a pot and leaves). Please refer to Fig A21 in our revised pdf.
>
> > [W3] visual quality is sometimes slightly lower than SOTA video generation models due to 3D reconstruction error.
>
> We acknowledge that in some cases, our visual quality is slightly lower than that of state-of-the-art video generation methods; we believe that employing a stronger reconstruction model or incorporating generative priors could alleviate this issue. We have updated the manuscript accordingly and added a more detailed discussion of this point in the Discussions section.
>
> > [Q1] How will you extend NGFF to robustly predict 4D dynamics from monocular or partial RGB inputs?
>
> Thank you for this valuable suggestion. We agree that extending NGFF to handle monocular or partial inputs is crucial in achieving human-like physical reasoning ability. In fact, our framework is not inherently limited to multi-view settings. Below, we clarify our current capabilities and outline how we can leverage state-of-the-art reconstruction methods to robustly extend NGFF to these challenging scenarios:
> 1. Current Capabilities on Partial/Monocular Inputs. We would like to clarify that partial RGB observations (e.g., occlusions) is already included in our current dataset.  To enhance the topological integrity of the object's 3D Gaussian representation, we introduced a refinement module using DiffSplat, a 3D asset generation model, after our feed-forward reconstruction. This module inherently enables our model to reconstruct objects from even monocular inputs and robustly predict subsequent dynamics. We added extra results using monocular RGB as input. Please refer to our revised manuscript 【Figure A20 in supplementary material】
> 2. While monocular inference is feasible, we want to explain the reason why we choose multi-view settings.
> - NGFF framework follows the standard practice adopted by the majority of contemporary 3D Gaussian and neural rendering systems. Such multi-view data are widely available in common benchmarks, and thus our main experiments are conducted under this widely used and well-supported setting.
> - In GSCollision dataset, monocular input introduces significant ambiguity in object depth and pose estimation. As observed in baseline failure cases (e.g., PhysGen3D), inaccurate geometry severely impacts downstream physical dynamics, such as collision handling. Therefore, multi-view data is currently necessary to guarantee the geometric fidelity required for physical simulation.
> 3. Extending roboustness via recent advances in 3D community. While our current model exhibits reduced fidelity when using monocular input, thanks to recent advances in powerful general-purpose 3D object reconstruction models, including [1]Trellis and [2]Hunyuan3D 2.5 for high-fidelity single-image object reconstruction, [3]Amodel3R and [4]CUPID for robust recovery from partial observations, and [5]Gen3C for scene-level generative reconstruction，the geometry and appearance estimated from monocular or incomplete inputs can be seamlessly integrated into our framework. These reconstructed scene can serve as valid inputs to NGFF, enabling our method to naturally extend to monocular or partial observation settings.
>
> [1] Structured 3D Latents for Scalable and Versatile 3D Generation
>
> [2] Hunyuan3D 2.5: Towards High-Fidelity 3D Assets Generation with Ultimate Details
>
> [3] Amodal3R: Amodal 3D Reconstruction from Occluded 2D Images
>
> [4] CUPID: Pose-Grounded Generative 3D Reconstruction from a Single Image
>
> [5] GEN3C: 3D-Informed World-Consistent Video Generation with Precise Camera Control

---

### Official Review · Reviewer_T5cC · 2025-11-03

**Soundness:** 2
**Presentation:** 3
**Contribution:** 2
**Rating:** 4
**Confidence:** 4

**Summary:**

This work presents the Neural Gaussian Force Field (NGFF) for animating 3D Gaussians. In summary, subjects are reconstructed from multi-view visual observations, and a continuous force field animates these reconstructed objects. The force field (NGFF) is predicted using a neural network. To train this force-prediction network, the authors simulate a large-scale dataset using the Moving Particle Method (MPM). The key contributions of this work are the development of the force-prediction network and the creation of the proposed dataset.

**Strengths:**

This work introduces a new approach to generating physically plausible 4D Gaussians in a feed-forward manner. The force-field prediction network is innovative.

The motivation and method are clearly presented, making them easy to understand.

The proposed dataset is valuable for studying 4D physics-plausible Gaussians.

**Weaknesses:**

I have two concerns on this work.

I have two concerns regarding this work:

1. The quality of the proposed dataset:
   (1) Is the MPM simulator truly representative of real-world physics? I worry that a simple MPM may not produce high-quality physics data. If that’s the case, the dataset's value may be diminished.
   (2) The simulations are conducted in a simple box environment, which lacks complexity and reduces diversity.
   (3) Although 3DGS offers high fidelity as a representation, there is still a noticeable gap between the rendered video and real-world captured video, as illustrated in Fig. 3.

2. The soundness of the force field prediction network:
   (1) The predicted target is highly dimensional and ambiguous. I doubt whether the model can effectively learn PDE dynamics from the data.

**Questions:**

- The author should provide clearer information about the dataset: (1) How many particles are included? (2) How are collisions handled? (3) Which constitutive model is used in the Material Point Method (MPM)? (4) How many simulation steps were performed?

- How can you demonstrate that the predicted results align with PDE knowledge rather than relying solely on mass dynamics? (This is an open question.)

- Have you considered objects with uneven quality distribution?

- If the surface of the environment is more complex, such as featuring uneven slopes, can your model accommodate this?

---

> ### Author Response · Authors · 2025-11-21
> **Response to reviewer T5cC (Part 1)**
>
> > [W1] The quality of the proposed dataset
>
> We appreciate the concerns regarding the dataset's quality and the fidelity gap between simulated and real-world data.
> Collecting and processing real-world videos into trainable components, such as 3D representations with point correspondence, presents a significant, **costly scalability barrier**. Therefore, the primary motivation for employing MPM simulation was to enable **easy, fast, and cost-effective** generation of physics-grounded 4D data at scale. This paradigm is crucial for research in dynamic prediction and robotics, where models are often trained in simulation before transfer to the real world.
>
> To ensure the highest possible fidelity, we utilized an advanced GPU-accelerated MPM engine, featuring **high-resolution** grids and simulation steps, along with objects and backgrounds **reconstructed from real-world scenes** (e.g., WildRGBD). For the first step, we focus on complex object interactions on a table, so we choose a box area for simulation. While this pipeline may not eliminate the sim-to-real gap entirely, it provides a highly scalable and flexible platform for rapidly validating the effectiveness of our method. We believe further improvement can be achieved with the progress in both physics simulation (MPM) and 3D reconstruction techniques (3DGS), and future work will focus on further enhancing the quality of the simulated data.
>
> Besides, another reasonable next step to alleviate sim-to-real gap is to study how to transfer the simulation data into the real-world by fine-tuning the pretrained model on a small set of real data.
>
> > [W2] The soundness of the force field prediction network
>
> Thank you for your feedback. Instead of directly predicting ambiguous, high-dimensional particle changes, we constrain the learning space by decomposing the dynamics into simpler targets: **Global Transformation Force Field** and **Local Stress Field**. We model global rigid transformation as a 6D vector capturing overall object translation and rotation, derived from a neural operator on a relational graph. We model local deformation as a point-wise field only where needed (e.g., contact regions), reducing the ambiguity. A second-order **Ordinary Differential Equation (ODE)** solver then stably integrates the force field to generate continuous, physically plausible 4D trajectories.
>
> The results in Table 1 shows that this explicit, physics-grounded formulation allows NGFF to achieve strong **out-of-distribution (OOD)** generalization across spatial, temporal, and compositional shifts, outperforming sequence models. The ablation results in the following table suggest that the model indeed learns PDE-related knowledge beyond simple rigid dynamics, as the performance drops if we don't consider the deformation modeling.
>
> | Model | Spatial RMSE ↓ | Temporal RMSE ↓ | Compositional RMSE ↓ |
> | :--- | :--- | :--- | :--- |
> | NGFF w/o deformation | 0.110 | 0.144 | 0.131 |
> | **Our NGFF** | **0.082** | **0.107** | **0.104** |
>
> > [Q1] The author should provide clearer information about the dataset: (1) How many particles are included? (2) How are collisions handled? (3) Which constitutive model is used in the Material Point Method (MPM)? (4) How many simulation steps were performed?
>
> Thank you for your suggestion. We have added these details to our revision (Appendix A.1 Data generation).
> 1. We list out the number of particles for each object below.
> | Object | ball | bowl | can | cloth | duck | miku | panda | phone | pillow | rope |
> | :--- | :--- | :--- | :--- | :--- | :--- | :--- | :--- | :--- | :--- | :--- |
> | **Particles** | 106115 | 232764 | 70569 | 116188 | 80544 | 48316 | 56104 | 26571 | 64229 | 8051 |
> 2. In our system, collisions are handled entirely at the grid stage following standard practice in modern MPM frameworks (e.g., ChainQueen, NVIDIA Warp and DiffTaichi). The simulation proceeds in a P2G → Grid Update → Collision (Grid BC) → G2P pipeline. Collision detection and response are applied after the grid momentum update and before transferring velocities back to particles.
> Each collider (plane, cuboid, bounding box, etc.) registers a boundary-condition kernel. During each step, the simulator iterates through these kernels and evaluates whether each grid node lies inside the collider region. When a grid node is detected to be inside (or crossing) a collider, we apply a velocity projection consistent with the collider's physical model. These boundary conditions are imposed directly on the grid velocity field. After enforcing the grid boundary conditions, particles gather updated grid velocities during the G2P stage. Because particle velocities are derived from these corrected grid velocities, particles naturally satisfy non-penetration and frictional constraints without explicit particle-level collision detection.
> 3. We utilized Neo-Hookean elasticity as the constitutive model.
> 4. We simulate each scene for 100 main steps, corresponding to a total of 20,000 substeps.

---

> > ### Author Response · Authors · 2025-11-21
> > **Response to reviewer T5cC (Part 2)**
> >
> > > [Q2] How can you demonstrate that the predicted results align with PDE knowledge rather than relying solely on mass dynamics?
> >
> > That’s an insightful question. To test whether our model truly captures the PDE component rather than merely learning mass dynamics, we performed an ablation where the deformation modeling was removed. The resulting performance degradation suggests that the model indeed learns PDE-related knowledge beyond simple mass dynamics.
> >
> > | Model | Spatial RMSE ↓ | Temporal RMSE ↓ | Compositional RMSE ↓ |
> > | :--- | :--- | :--- | :--- |
> > | NGFF w/o deformation | 0.110 | 0.144 | 0.131 |
> > | **Our NGFF** | **0.082** | **0.107** | **0.104** |
> >
> > > [Q3] Have you considered objects with uneven quality distribution?
> >
> > Thank you for your suggestion. We found the term "quality distribution" a bit confusing, and for now, we assume you meant "mass distribution." If you were actually referring to something else, such as visual quality, we are happy to provide further clarification.
> > We have added an experiment to model an object with non-uniform mass distribution (a shaking ficus consisting of a pot and leaves). The results in our revised pdf (Figure A21) show that our model can learn the dynamic pattern of non-uniform objects with a small MSE loss of 2.54e-4.
> >
> > > [Q4] If the surface of the environment is more complex, such as featuring uneven slopes, can your model accommodate this?
> >
> > Yes. In our framework, environment surfaces are treated as boundary conditions and provided as model inputs. This enables us to detect boundary collisions and predict boundary forces using a dedicated network. For uneven slopes, we can apply the same strategy: perform boundary collision detection and train a network to learn the resulting boundary forces, without modifying the overall model architecture.

---

### Meta-Review · Area_Chair_a1hp · 2025-12-08

**Summary:**

While the paper is generally well-written and the proposed NGFF framework is viewed as innovative, reviewers raised substantial concerns about
1. reliance on synthetic MPM data and resulting limits on realism,
2. under-determined or unclear aspects of the force-field formulation,
3. limited generalization beyond the 10 predefined objects,
4. reconstruction-module inconsistencies, and
5. scalability and data-efficiency claims.

After rebuttal, three reviewers still see core conceptual issues unresolved, and one reviewer (QtAo) becomes more negative.

**Reviewer Concerns:**

**Addressed**

- Dataset details and simulation settings (T5cC).
- Clarification of reconstruction options and monocular/partial-input behavior (jh1Y).
- Implementation specifics (losses, supervised Gaussian attributes, Eq.6) (sf96).
- Terminology clarification and removal of photometric loss (QtAo).

**Outstanding**

- Realism and fidelity limitations of MPM supervision (T5cC, sf96).
- Whether NGFF meaningfully generalizes to unseen objects or unseen object combinations (sf96, QtAo).
- Scaling and data-efficiency doubts (4TB / 10 objects) (QtAo).
- Under-determination of local force-field prediction (QtAo).
- Reconstruction-module inconsistency and unclear role in the full pipeline (QtAo).
- Overstatement of photorealism and video-generation competence (QtAo).

These outstanding points concern fundamental claims of the paper and were not resolved by rebuttal.

**Reviewer Scores:**

1. **T5cC: 4 → 4**: The rebuttal clarifies technical details but does not resolve core concerns about MPM fidelity, physical soundness, and generalization to complex settings, so the reviewer would likely maintain the original score.
2. **jh1Y: 8 → 8**: This reviewer was already strongly positive, and their limited concerns were adequately addressed, leaving no reason for either an increase or decrease in score.
3. **sf96: 6 → 6**: Although implementation details were clarified, the fundamental issues—dependence on MPM, limited evidence of generalization to unseen physics, and uncertainty about real-data training—remain unresolved, so the score would stay unchanged.
4. **QtAo: 6 → 4**: The rebuttal did not alleviate this reviewer’s major criticisms; instead, their follow-up response became more negative regarding generalization, reconstruction consistency, scalability, and formulation soundness, likely leading to a lower score.

---

### Decision · Program_Chairs · 2026-01-26

Accept (Poster)